# Natural optical activity as the origin of the large chiroptical properties in π-conjugated polymer thin films

Jessica Wade[1,2], James N. Hilfiker [3], Jochen R. Brandt [2,4], Letizia Liirò-Peluso [5,6], Li Wan [1,2], Xingyuan Shi [2,4], Francesco Salerno[2,4], Seán T. J. Ryan[4], Stefan Schöche[3], Oriol Arteaga[7], Tamás Jávorfi[8], Giuliano Siligardi [8], Cheng Wang [9], David B. Amabilino [5], Peter H. Beton [6], Alasdair J. Campbell[1,2✉] & Matthew J. Fuchter [2,4✉]

Polymer thin films that emit and absorb circularly polarised light have been demonstrated with the promise of achieving important technological advances; from efficient, high-performance displays, to 3D imaging and all-organic spintronic devices. However, the origin of the large chiroptical effects in such films has, until now, remained elusive. We investigate the emergence of such phenomena in achiral polymers blended with a chiral small-molecule additive (1-aza[6]helicene) and intrinsically chiral-sidechain polymers using a combination of spectroscopic methods and structural probes. We show that – under conditions relevant for device fabrication – the large chiroptical effects are caused by magneto-electric coupling (natural optical activity), not structural chirality as previously assumed, and may occur because of local order in a cylinder blue phase-type organisation. This disruptive mechanistic insight into chiral polymer thin films will offer new approaches towards chiroptical materials development after almost three decades of research in this area.

[1] Department of Physics, Imperial College London, South Kensington Campus, London SW7 2AZ, UK. [2] Centre for Processable Electronics, Imperial College London, South Kensington Campus, London SW7 2AZ, UK. [3] J.A. Woollam Co. Inc., 645M Street, Suite 102, Lincoln, NE 68508-2243, USA. [4] Department of Chemistry and Molecular Sciences Research Hub, Imperial College London, White City Campus, 82 Wood Lane, London W12 0BZ, UK. [5] School of Chemistry & The GSK Carbon Neutral Laboratories for Sustainable Chemistry, University of Nottingham, Triumph Road, Nottingham NG7 2TU, UK. [6] School of Physics and Astronomy, University of Nottingham, University Park, Nottingham NG7 2RD, UK. [7] Departament de Física Aplicada, Universitat de Barcelona, IN2UB, Barcelona 08028, Spain. [8] Diamond Light Source, Harwell Science and Innovation Campus, Didcot, Oxfordshire OX11 0DE, UK. [9] Advanced Light Source, Lawrence Berkeley National Laboratory, Berkeley, California 94720, USA. ✉email: alasdair.campbell@imperial.ac.uk; m.fuchter@imperial.ac.uk

Chirality is a fundamental symmetry property that is present in all natural and life sciences, from molluscs to peptides, small molecules to spiral galaxies. Just as molecules can exist in left-handed (LH) and right-handed (RH) mirror image pairs, light can feature either LH or RH circular polarisation, depending on the sense of rotation of the electric field vector with respect to the direction of propagation. The manipulation of such (chiral) circularly polarised light (CPL) has received widespread attention over the past decades as it presents opportunities in next-generation optoelectronics; through, for example, the development of organic chiral semiconducting materials[1–6]. The tunable electronic properties, ability to absorb and emit CPL, and potential for spin-polarised electron transport in chiral organic semiconducting materials promises far-reaching applications, including in enantioselective sensing, display technologies and quantum computation[7–14]. A number of methods have been explored to create chiroptically active solid-state structures, including the use of chiral solvents, the incorporation of chiral sidechains into otherwise achiral polymers, and the combination of an achiral polymer with a chiral small-molecule additive[2,15–19]. The broadly held understanding is that such materials exhibit chiroptical effects in thin films that originate at the local/molecular level (i.e., natural optical activity, dissymmetry $\approx 10^{-3}$), through the presence of longer-scale structural chirality (i.e., a periodic helicoidal structure as in a cholesteric-like phase, dissymmetry $\approx >0.1$), or a combination of the two[20].

Chiroptical properties for chiral sidechain polymers, which are by far the most explored materials of this class, are typically attributed to the supramolecular self-assembly of polymer chains in the solid state[2,21–25]. A nematic chiral helicoidal (cholesteric-like) packing structure, which is typically characterised through the use of an alignment layer, has been used to account for 1) the dissymmetry of absorption and 2) luminescence of circularly polarised light (CPL) in almost all polymer systems (Supplementary Table 1) and how these photophysical values vary with properties such as film thickness. The exact pitch length of such helical assemblies varies depending on the chemical composition, fabrication technique, and experimental approach, and could be as tight as 300 nm (chiral poly{9,9-bis[(3 S)-3,7-dimethyloctyl}-2,7-fluorene), cPFO)[26] or as wide as 1500 nm (chiral poly(9,9′-dialkylfluorene-alt-2,5-dialkoxyphenyl) copolymers)[2]. For mono- or multi-domain cholesteric films with such pitch lengths, film thicknesses of >500 nm would be required to achieve high dissymmetry (g-factors > 1) in circularly polarised (CP)-absorption and emission from the polymer via mechanisms such as Bragg reflection. However, such thick structures are generally not compatible with the majority of device applications (active layer ≤ 200 nm), where increasing film thickness typically diminishes device performance, other than in rare examples[27–29].

We and others have reported an alternative approach to generate high dissymmetry CPL in devices using materials where an achiral polymer is blended with a chiral additive (ACPCA hereafter)[17,30,31]. These systems can notably achieve exceptionally high dissymmetry (absolute absorption and emission g-factors ≈ 0.25–1.5) at film thicknesses that are compatible with highly efficient device applications (~150 nm) and without the use of an alignment layer. The combination of high dissymmetry at low thicknesses is inconsistent with the optical mechanisms that would occur in structurally chiral systems, such as those that have been proposed for cholesteric chiral sidechain polymers (CSCP) or ACPCA films on alignment layers[22,30–34]. It is evident that a new model is required to describe these systems and fully realise the application of polymer thin films in efficient devices that make use of circularly polarised light.

In this work we perform a systematic investigation to establish the origins of the large chiroptical effects in chiral polymer systems, with and without alignment layers. To understand the role of alignment layers, polymer chemical structure, and post-deposition treatment, we combine our previous chiral small-molecule additive — enantiopure aza[6]helicene ([M] or [P] aza [6]H) — with three achiral polyfluorene-based polymers: poly (9,9-dioctylfluorene) (PFO), poly(9,9-dioctylfluorene-alt-benzothiadiazole) (F8BT), and poly(9,9-dioctylfluorene-alt-bithiophene) (F8T2). We then compare such blends to polyfluorene derivatives bearing chiral sidechains (cPFO and cPFBT, see Fig. 1 for structures and Supplementary Methods for further details)[35]. We employ Mueller matrix spectroscopic ellipsometry (MMSE) to create the first optical model to describe the electromagnetic interactions of non-aligned ACPCA thin films. The results show that for non-aligned thin films of both CSCP and ACPCA systems, the chiroptical effects do not originate from the mechanism involving solely a twisted dielectric tensor, such as would be observed in a structurally-chiral cholesteric liquid crystal phase (long-range effects). Instead, these effects occur due to so-called natural optical activity, an effect based on short-range interactions between magnetic and electric dipoles[20]. We find that it is the use of an alignment layer—something that is rarely employed in CP-OLED devices due to their insulating properties—that results in the formation of a chiral structurally ordered mesophase. Resonant soft x-ray scattering (RSoXS) and atomic force microscopy (AFM) suggest that the natural optical activity may arise due to the assembly of twisted polymer fibrils into a double twist cylinder type blue phase. This understanding of chiral semiconducting polymer systems will help optimise their properties towards ultrathin, stretchable and efficient CP sensitive/emissive devices for a range of technologies.

## Results

**Chiroptical response of ACPCA and CSCP films.** Following our previous optimisation of non-aligned F8BT-based ACPCA thin films[30], we broadened this material class by introducing 10 wt% of the chiral small-molecule additive 1-aza[6]helicene (aza[6]H) to three different polymers, F8BT, PFO and F8T2, in solution (Fig. 1 for structures). These blend ACPCA materials and two CSCP polymers (cPFO and cPFBT) were subsequently spin-coated into thin films, annealed at 140 °C for 10 min in a $N_2$ glovebox and quickly cooled to room temperature. In their initial as-cast state, as shown in Fig. 2, the PFO- and F8BT-based ACPCA films are virtually Circular Dichroism (CD)-silent in the absorption bands of the polymers (between 390 and 500 nm), but contain the CD signature of the aza[6]H (≈334 nm). The F8T2 thin films show a weak Cotton effect (≈30 mdeg, λ = 498 nm) near their lowest energy transition (inset of Fig. 2). On the other hand, as-cast CSCP thin films have non-negligible CD, with cPFBT demonstrating a pronounced CD signal (>600 mdeg, λ = 470 nm) (Fig. 2) even before annealing. In all cases, annealing dramatically increases the chiroptical activity of the lowest energy transition (PFO: 3,110 mdeg, F8BT: 10,200 mdeg, F8T2: 21,500 mdeg, cPFO: 13,900 mdeg, cPFBT: 12,100 mdeg, see Supplementary Table 2 and Supplementary Fig. 1). High resolution (50–100 μm) spatially resolved CD imaging confirms that this chiroptical response is remarkably uniform over large areas (Supplementary Fig 2)[36,37]. As has been reported elsewhere, the cross-coupling of linear birefringence (LB) and linear dichroism (LD) and CP-selective reflection from structurally-chiral cholesteric stacks can result in apparent optical activity[18,38]. For example, in liquid crystalline materials that are configured in a nematic chiral helicoidal structure, the pitch can be tuned such that the modulation of refractive index through a given thickness

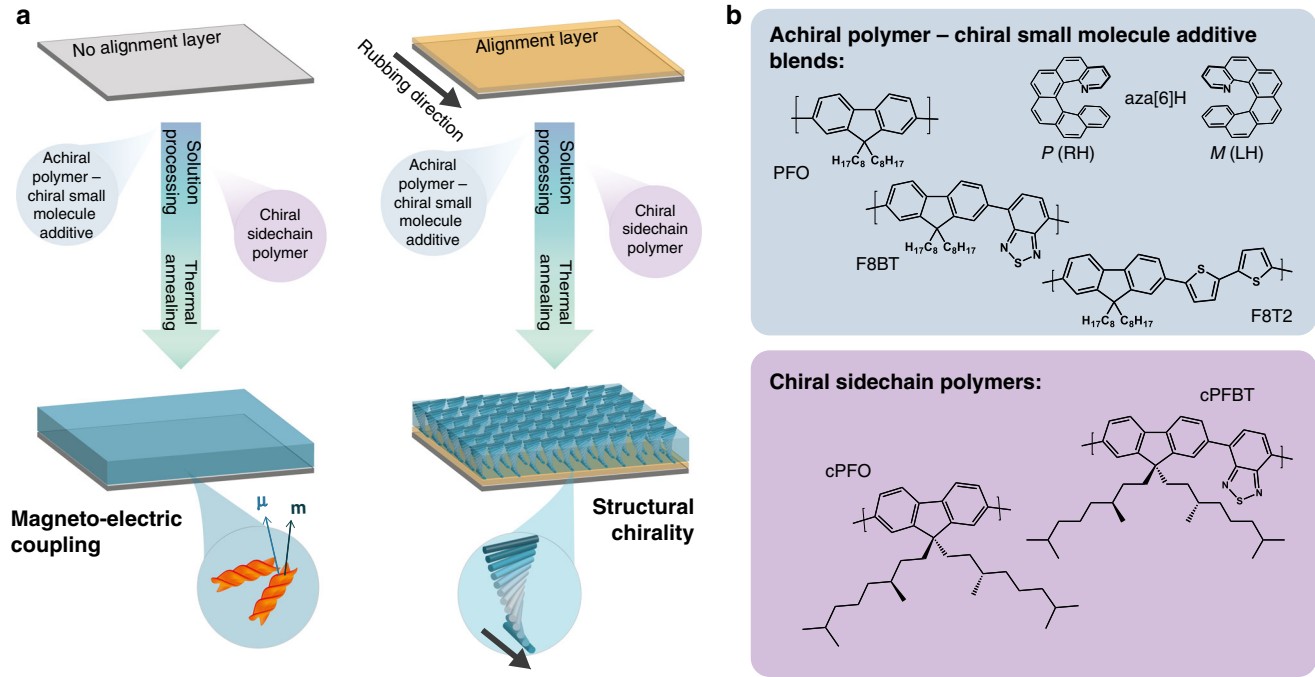

**Fig. 1 Origins of the chiroptical effects explored in this work.** (**a**) a cartoon depicting the mechanisms that underpin the chiroptical response in non-aligned and aligned thin films and (**b**) the polymer systems evaluated here.

produces an apparent CD due to circular birefringence. It is not possible, however, to discriminate between specific origins of such chiroptical activity – natural optical activity or structural chirality – just using steady-state transmission based measurements such as CD[20]. Monodomain cholesteric stacks (i.e., structural chirality that is out of the substrate plane) with a given pitch demonstrate temperature-sensitive Bragg reflection at a particular wavelength (the photonic band gap), which is not observed in systems with purely natural optical activity. However, Bragg reflection characteristics of a cholesteric stack are rarely observed in non-aligned polymer thin films[17,21]. To overcome potential artefacts and establish the true origins of the chiroptical effect in polymer thin films, we have performed detailed MMSE in both transmission and reflection, which is well suited to the optical analysis of thin films of complicated chiral systems[39]. MMSE has previously been used to evaluate the structural CD and Circular Birefringence (CB) in cellulose, small-molecule aggregates and non-conjugated polymers[22,40–42]. Further mathematical descriptions of the MMSE are provided in the Supplementary Discussion 1.

Annealed thin films of the neat achiral polymers (F8T2, F8BT, PFO) (Supplementary Fig. 3) exhibit uniaxial anisotropy (of the form in Supplementary Eq. 2) with the optical axis perpendicular to the sample surface. Uniaxial anisotropy of this nature can be described by diagonal dielectric tensor elements (Supplementary Eq. 4), the individual terms of which are illustrated in Supplementary Fig. 4. In contrast to the annealed films of neat polymers, annealed, non-aligned ACPCA and CSCP films (Fig. 3, Supplementary Figs. 5–6) exhibit both CD and CB, (as inferred from matrix elements $M_{14}$, $M_{23}$, $M_{32}$, $M_{41}$,) in transmission. These circular elements do not vary when the sample is rotated and are silent in reflection for these films.

**Optical models generated to interpret the chiroptical response.** We first attempted to model the MMSE data by assuming the chiroptical effects arose from a either a mono- or multi-domain cholesteric stack structure. As the refractive index of these materials is ≈2 at wavelengths close to the maxima of the lowest energy CD band, the pitch of a mono-domain cholesteric would need to be ≈250 nm to generate the chiroptical effects observed (Supplementary Fig. 7). In transmission, this model of structural chirality produces linear effects in several MM elements that are not exhibited in experimental data (Supplementary Fig. 7), whilst in reflection, the monodomain cholesteric stack model produces strong differential reflection of CPL that does not appear in our measurements (Fig. 3, red curves). These findings show that the origins of the CB and CD in non-aligned chiral films do not result from a mesoscopic model based on a dielectric tensor that is a periodic function of the thickness, as would occur in structurally chiral monodomain cholesteric stack systems. We shall refer to this mesoscopic description as the twisted dielectric tensor model. We next considered helical multi-domain models[43], which evaluate the optical response of incoherent (domains > the wavelength of light) and coherent (domains ≤ the wavelength of light) superpositions of cholesteric grains (Supplementary Discussion 2 and Supplementary Fig. 8). Neither an incoherent or coherent multi-domain model can satisfy the MMSE results acquired at multiple angles of incidence or without introducing significant depolarisation, which is not observed in our measurements. The uniformity of the circular response was evaluated using spatially resolved MMSE (Supplementary Fig. 8), which, comparable to the spatially resolved CD measurements described above (Supplementary Fig. 2), show no evidence of a mono or multi-domain structure.

Instead, to describe the optical activity recorded in transmission and reflection of the non-aligned ACPCA and CSCP thin films requires them to be modelled as a bianisotropic medium. Just as with the neat polymers, uniaxial anisotropy is required to fit the linear terms of the MM, whereas the circular terms require magneto-electric coupling that is represented by an optical activity tensor. The tensorial constitutive equations for the ACPCA and CSCP thin films can be written in the following

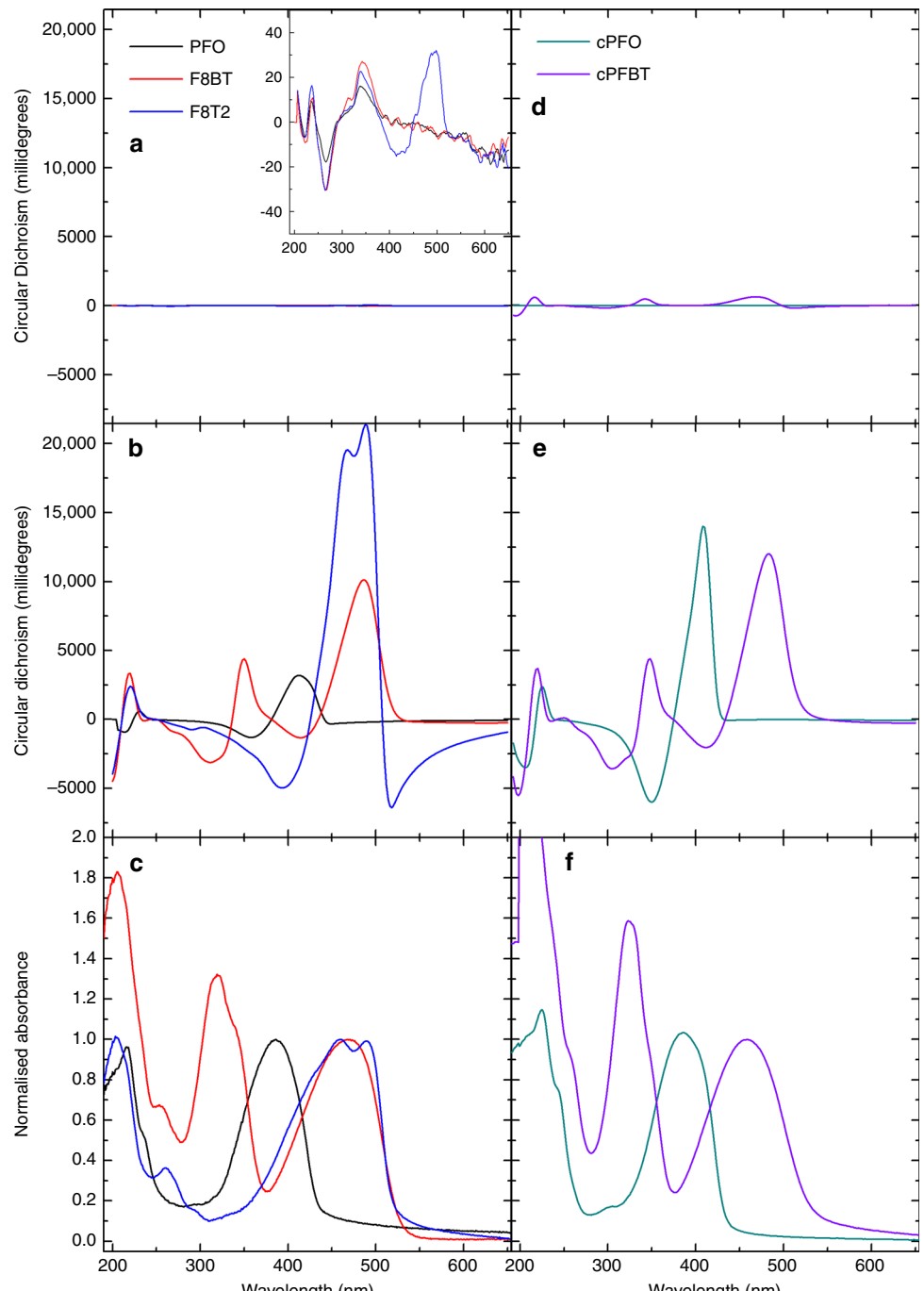

**Fig. 2 Absorbance and circular dichroism (CD) of 130 nm ACPCA and CSCP thin films.** CD of ACPCA (*P*- aza[6]H) (PFO: black, F8BT: red and F8T2: blue lines) and CSCP (cPFO: turquoise, cPFBT: purple lines) thin films measured (**a**, **d**) before and (**b**, **e**\*) after annealing for 10 min at 140 °C and being removed from the heat source. The absorption spectra, recorded after annealing, normalised to the lowest energy transition, are also provided (**c**, **f**). The inset of (**a**) shows a 500-fold zoom of the CD spectrum, revealing the CD signature of the trace additive aza[6]H in the c. 200–400 nm region. \*To allow for direct comparison between the shape and relative intensities of absorption bands (**f**), the CD spectra in (**e**) represent the experimental data reflected over the x-axis.

compact form (Eq. 1)[44].

$$\begin{bmatrix} D \\ B \end{bmatrix} = \begin{bmatrix} \varepsilon & i\alpha \\ -i\alpha^T & \mu \end{bmatrix} \begin{bmatrix} E \\ H \end{bmatrix} \qquad (1)$$

Here $E$ refers to the electric field strength, $B$ the magnetic flux density, $D$ the electric displacement density and $H$ the magnetic field strength, $\varepsilon$ refers to the permittivity tensor, $\mu$ is the permeability tensor (for nonmagnetic media it is the $3 \times 3$ identity) and $\alpha$ is the optical activity tensor (Supplementary Discussion 1). The explicit $6 \times 6$ tensor that has been used to fit all

experimental data is given by $\boldsymbol{A}$ (Eq. 2).

$$\boldsymbol{A} = \begin{bmatrix} \varepsilon_x & 0 & 0 & i\alpha_x & 0 & 0 \\ 0 & \varepsilon_x & 0 & 0 & i\alpha_x & 0 \\ 0 & 0 & \varepsilon_z & 0 & 0 & i\alpha_z \\ -i\alpha_x & 0 & 0 & 1 & 0 & 0 \\ 0 & -i\alpha_x & 0 & 0 & 1 & 0 \\ 0 & 0 & -i\alpha_z & 0 & 0 & 1 \end{bmatrix} \qquad (2)$$

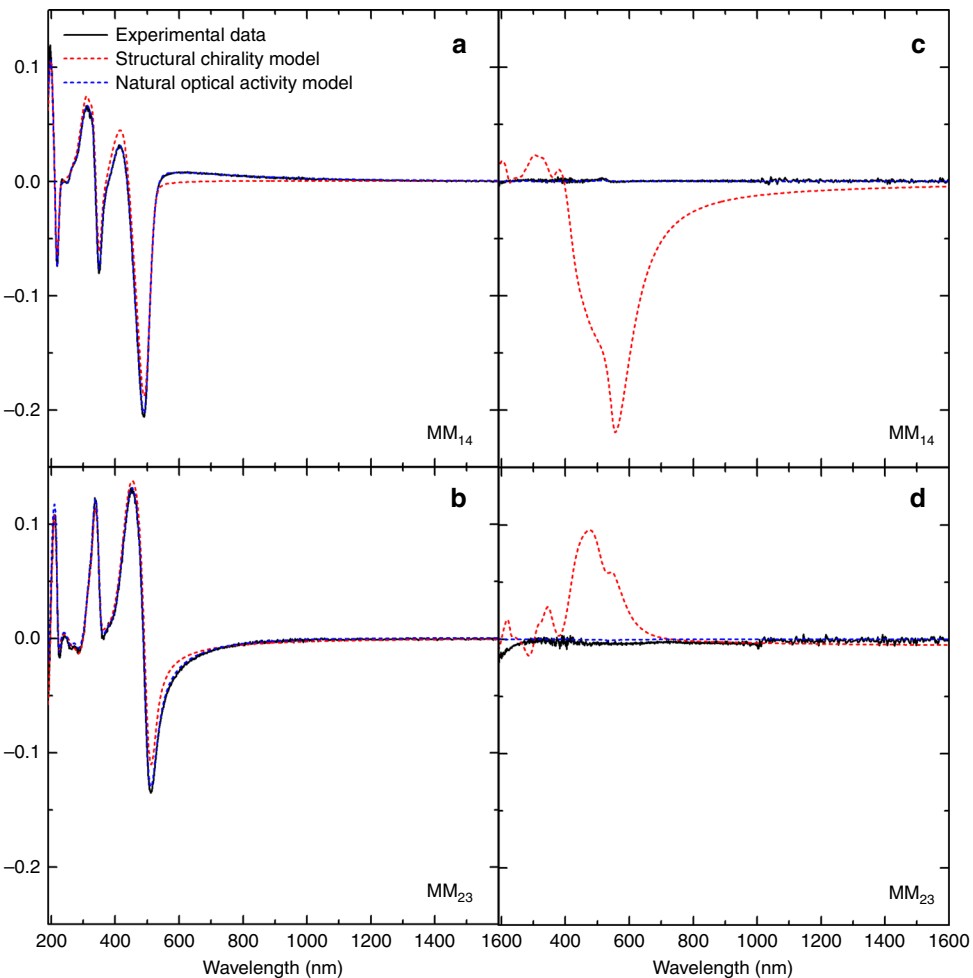

**Fig. 3 Experimental and modelled MM data for non-aligned ACPCA thin films.** Measured MM data (black solid line) associated to CD ($M_{14}$) and CB ($M_{23}$) recorded in transmission (**a**, **b**) and reflection (**c**, **d**) along with a model-calculated match to the MM transmission and reflection data for a 131 nm thick non-aligned F8BT:aza[*M*]H film assuming the origins of the chiroptical effect are structural chirality (red dashed line) or natural optical activity (blue dashed line).

While the twisted dielectric tensor model does not fit the obtained MM data, magneto-electric coupling given by the optical activity tensor (Eq. 2), can simultaneously predict the transmitted and reflected MM in all our chiral polymeric systems (Fig. 3). The optical activity tensors that we have determined for all three ACPCA systems are respectively shown in Supplementary Fig. 9 and require the same uniaxial symmetry as the dielectric tensor of neat polymers.

In an effort to rationalise this observation with previously reported results, we introduced anisotropic alignment layers of rubbed polyimide (PI) before spin-coating the polymer thin films[17,45,46]. Rubbed PI layers have been shown to promote the unidirectional alignment of liquid crystalline materials[46,47]. In contrast to their non-aligned counterparts, the MMSE of polymer thin films with alignment layers have linear and circular terms in both reflection and transmission (Supplementary Fig. 10), which cannot be isolated upon sample rotation. For the aligned CSCP films we first attempted to fit these data using an optical model with natural optical activity but found that it could not accurately reproduce both linear and circular terms simultaneously (Fig. 4, Supplementary Fig. 11). In contrast, when we introduced the twisted dielectric tensor model (considering a helical dependence in ε, as one would expect for a cholesteric stack) (Fig. 4, Supplementary Fig. 12) we could qualitatively fit both the reflected and transmitted data at a range of incident angles using

a twist of ≈85°. Whilst the same appears to be true for the aligned ACPCA films, there seems to be a combination of natural optical activity and a twisted structure for such materials, which complicates our analysis (Supplementary Fig. 13).

**Morphological investigations.** In order to obtain structural insight into the features that underpin the optical properties, the molecular conformation of the chiral phase in the annealed, non-aligned neat and ACPCA films was probed using RSoXS at the carbon *K*-edge (E = 283.5 eV). RSoXS permits investigations into the orientation of carbon bonds in nanoscale helical assemblies, and is sensitive to molecular conformation in the plane of the substrate when recorded in transmission[48,49]. Whilst there is no clear scattering for the as-cast and annealed neat achiral polymer films, or for as-cast ACPCA thin films, the annealed ACPCA films show distinct scattering arcs, corresponding to an in-plane periodic feature with a characteristic length scale of 260–330 nm (Supplementary Figs. 14–15). For the annealed ACPCA thin films, there is a consistent offset between the incident beam direction and the scattering patterns, in opposite directions for [*P*] and [*M*] aza[6]H (Supplementary Fig. 14). The surface molecular packing of these systems was investigated using tapping-mode (noncontact AC mode) AFM. The measurements reveal that the annealed ACPCA films (Fig. 5, Supplementary

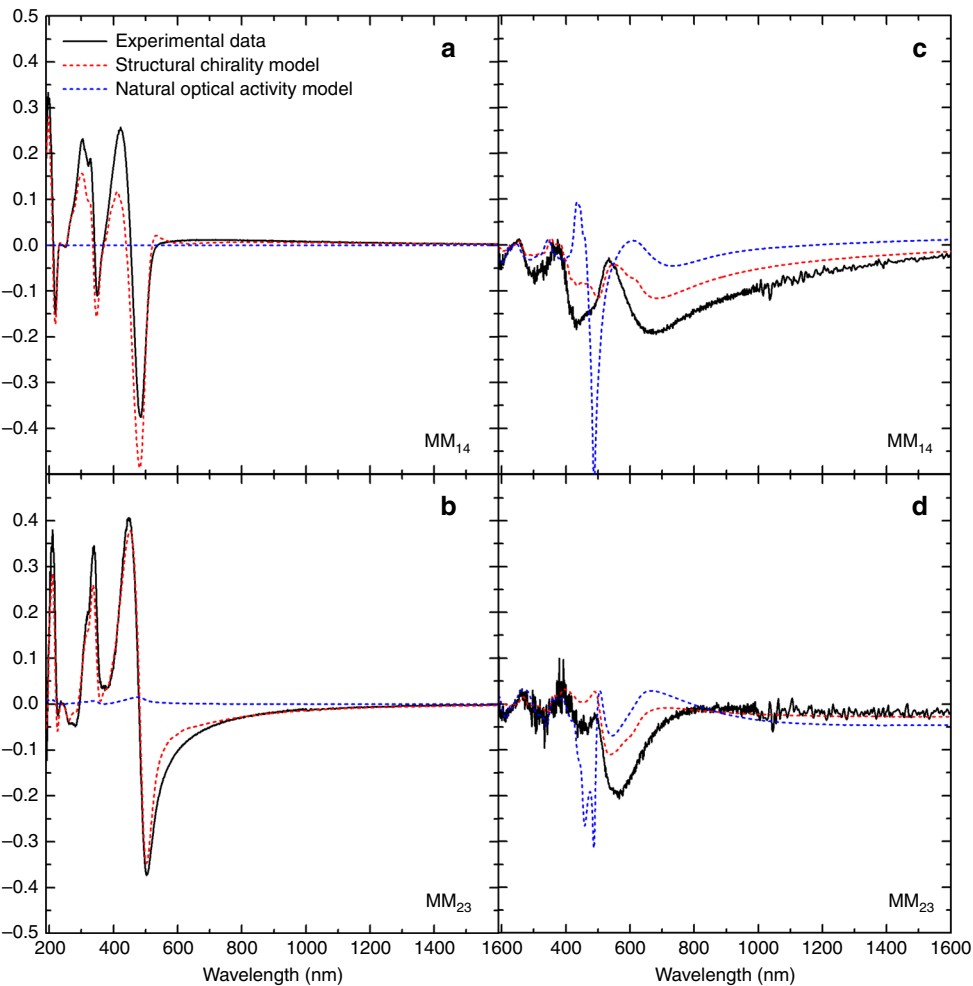

**Fig. 4 Experimental and modelled MM data for aligned CSCP thin films.** Measured MMSE data (black solid line) associated to CD ($M_{14}$) and CB ($M_{23}$) recorded in transmission (**a**, **b**) and reflection (**c**, **d**) along with a model-calculated match for an aligned cPFBT film assuming the origins of the chiroptical effect are structural chirality (red dashed line) or natural optical activity (blue dashed line).

Figs. 16–17) are very smooth compared to those of the neat polymer (RMS roughness, F8BT:aza[6]H: 0.49 nm, F8BT: 0.81 nm, calculated over a 1 μm² scan area). Whilst both systems adopt an isotropic fibril-like organisation, this pattern is more distinct for the annealed ACPCA thin films. The fibril domains (for the neat and ACPCA samples) are sub-micron in size, as revealed in the polarised optical micrographs of the films (Supplementary Fig. 17). The weakly birefringent ACPCA films (Supplementary Fig. 5) appear dark green-blue in transmission with crossed polarizers, and have a texture that is seen most clearly by applying a Berek compensator. In these samples, fibrils of various lengths diverge around features reminiscent of topological defects. Further details of the RSoXS and AFM measurements are reported in the Supplementary Information (Supplementary Figs. 14–17, Supplementary Discussion 3).

The AFM and RSoXS data appear to be consistent with the formation of a double twist cylinder blue phase[50,51]. This phase comprises a number of twisted polymer chains that form linear fibrils parallel to the substrate on the surface, which, in turn, are organised in cylinders that pack perpendicular to one another in the bulk (Fig. 6). The apparent topological defects in the AFM images are attributed to points where twisted fibrils (characteristic width of 25 nm, Supplementary Fig. 16) comprising of a few polymer chains with a variable-length emerge from the lower layers of the film. The aggregation of these fibrils would generate cylinders, spaced between 260 and 330 nm apart, a dimension observed with RSoXS. The tilt angle in the RSoXS pattern (Supplementary Fig. 14) indicates that these cylinders are comprised of double twisted polymer fibrils, and the lack of a distinct second-order Bragg peak (Supplementary Fig. 15) indicates that these cylinders have a weak structural order over large length scales. Double twist cylinder blue phases that pack into disordered structures have previously been observed in chiral liquid crystalline materials[51]. In the AFM images, the regions between the tops of the cylinders (topological defects), which would be disordered in the bulk of the sample in the blue phase model, comprise of pseudo-aligned fibrils because they are restricted to the surface plane (Fig. 5)[52]. The crossed polarised optical images of annealed ACPCA thin films recorded through a Berek compensator are comprised of interlaced finger-like domains, which is consistent with the formation of a blue phase[53]. The domain sizes are considerably smaller than those reported for small-molecule based systems. We attribute this to the fact that small-molecule systems are closer to equilibrium and will undergo Ostwald ripening faster into large domains during annealing.

As the intensity (Fig. 2) and origins (Fig. 3) of the chiroptical effects in annealed non-aligned ACPCA and CSCP thin films appear to be the same, despite different molecular strategies – chiral additive *vs.* chiral sidechain to induce a chiral film structure, we further compared these approaches using in situ CD mapping (Fig. 7, Supplementary Figs. 18–24). Heating both

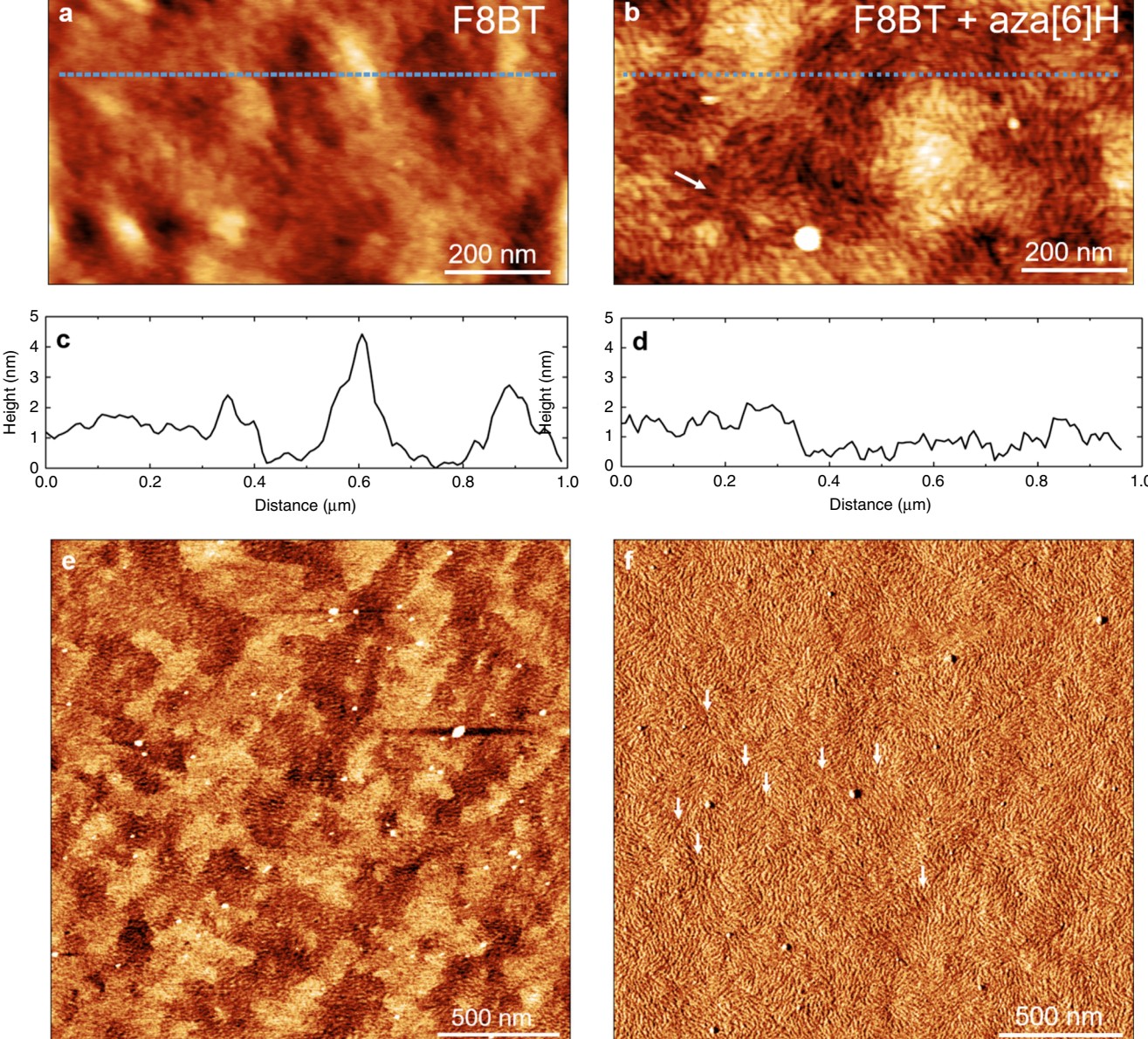

**Fig. 5 AFM images of annealed ACP and ACPCA thin films.** Topographic AFM images of (**a**) F8BT and (**b**) F8BT:aza[6]H annealed films; corresponding height profiles (**c**, **d**) extracted from the upper blue dashed trajectories in (**a**) and (**b**). Phase AFM images of (**e**) F8BT and (**f**) F8BT + aza[6]H annealed films. Images (**b**) and (**f**) show topological defects highlighted with white arrows.

the ACPCA and CSCP thin films causes an increase in the chiroptical activity of the lowest energy transition associated with the polymer, although with different kinetics (Fig. 7e, f)). For the ACPCA thin films, the onset of CD is consistent with the glass transition temperature of the given polymer. In contrast to the ACPCA thin films, where the temperature which gives rise to the strongest chiroptical signal, $T_{CD\ Max}$ (PFO: 160 °C, F8BT: 125 °C, F8T2: 120 °C), is sensitive to the chemical structure of the polymer, $T_{CD\ Max}$ is the same for both the CSCP thin films (cPFO: 140 °C, cF8BT: 140 °C) (Fig. 7d), Supplementary Fig. 23), above which there is considerable CD (≈11,000 mdeg). Further discussion relating to the phase behaviour of the thin films can be found in Supplementary Discussions 4 and 5. Taken together, this data suggests that for ACPCA materials it is essential to kinetically trap the chiral film structure formed around the glass transition temperature, whereas for CSCPs the structure induced by the chiral sidechains is more persistent over repeated heating-cooling cycles.

To further confirm that it is natural optical activity and not structural chirality that gives rise to the strong chiroptical response of non-aligned chiral polymer films, we measured the g-factors of aligned and non-aligned thin films of different thicknesses (Supplementary Fig. 25 and Supplementary Discussion 6). Once reflection losses are accounted for, the non-aligned thin films have thickness independent $g_{abs}$, as expected from natural optical activity, whereas aligned thin films show a $g_{abs}$ that increases linearly as a function of film thickness[41].

## Discussion
In this study, we compare two approaches to the formation of chiral polymer thin films: a chiral additive or a chiral sidechain. While these materials have different phase behaviour and thermal stability (Supplementary Discussion 5), both allow for a complementary means to achieve chiral polymer films with very large chiroptical activity. The comparative MMSE data obtained in this study show that the previously assumed formation of cholesteric

stacks for these chiral polymer systems only occurs when alignment layers are employed. In systems without alignment layers, the MMSE data can instead be fitted to an optical activity tensor that represents magneto-electric coupling (Fig. 3, Eq. 2). These findings indicate that the strong chiroptical effect observed in

non-aligned chiral polymer thin films arises from natural optical activity as opposed to the mesoscopic structural chirality previously assumed. Moreover, the observed anisotropy in optical activity, i.e., the different values of optical activity for light propagating parallel or perpendicular to the sample surface as expressed by the optical activity tensor, suggests that not only inter-chain interactions, but also intra-chain interactions, play a role in the overall chiroptical response. According to the AFM and RSoXS data, the magneto-electric coupling in annealed ACPCA systems may have its origin in the formation of a double twist cylinder-type blue phase[50,51], which to the best of our knowledge has never been reported for conjugated polymer materials.

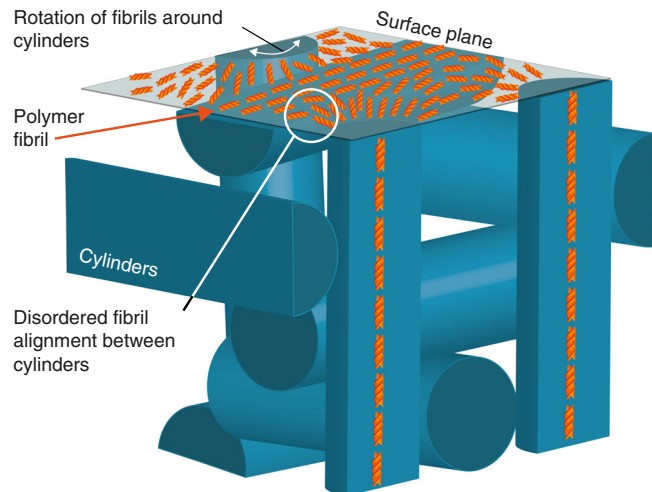

**Fig. 6 Schematic of the molecular model for non-aligned ACPCA thin films.** A representation of the proposed double twist cylinder blue phase in annealed, non-aligned ACPCA systems.

The very large chiroptical effects associated with a natural optical activity that we report contrast with the small effects typically observed for small molecules in solution, but is reminiscent of the high optical activity of chiral aggregates where supramolecular chirality leads to CD effects that are magnitudes higher than those of the isolated molecules[54–57]. Dissymmetry factors of isolated chromophores can be enhanced through the excitonic coupling of dipoles on nearby chromophores. Delocalisation of the excited state over multiple chromophores will also result in the breakdown of the dipole approximation, as the extended excited state is no longer significantly smaller than the wavelength of light. For unaligned ACPCA polymer films, the in-plane helical periodic modulation of the polymer in the double twist cylinder blue phase may give rise to a chromophore large enough to possess no local symmetry, meaning that transitions

**Fig. 7 In situ chiroptical response of ACPCA and CSCP thin films.** In situ CD spectra recorded during heating and cooling of (**a**) ACPCA (F8BT: aza[6]H) and (**b**) CSCP (cPFBT) thin films (note blue represents low temperatures and red represents high temperatures), (**c**) and (**d**) the CD intensity recorded at 480 nm as a function of temperature during heating (red) and cooling (blue), and (**e**) and (**f**) CD intensity of thin films held at 140 °C as a function of time for [P] (turquoise) and [M] (purple) systems (note the different time on-axis).

are electrically and magnetically allowed[58]. Exciton coupling of nearby polymer chains, as evidenced by the positive and negative Cotton bands in the CD spectra (Figs. 2 and 7), may serve to further enhance the chiroptical response, which leads to a greater contribution from the magnetic transition dipoles and electronic quadrupole transition to the CP dissymmetry.

It should be emphasised that the majority of applications of such materials in devices (for example CP-OLEDs) do not use alignment layers[21,30,33,59,60]. Therefore, assignment of molecular packing, and the chiroptical mechanisms that arise from this packing, which have been generated in aligned thin films cannot necessarily be valid in situations where such alignment has not been performed. Moreover, in devices, parameters other than the CP dissymmetry become important for optimal operation. Thick film structures are generally not compatible with the majority of device applications (active layer ≤ 200 nm), where increasing film thickness typically diminishes device performance. As a result, chiral materials that can generate large chiroptical responses in very thin films via natural optical activity have greater technological potential than those which require structural chirality. Furthermore, the large dissymmetry factors measured in transmission for aligned polymer thin films arise mainly due to reflection losses (i.e., the differential reflection of left- and right-handed light), whereas in the non-aligned polymer films the dissymmetry is the result of the intrinsic absorption of CPL. Such an outcome creates many other opportunities in the application of the aligned/non-aligned films. For example, isotropic non-aligned films demonstrating a large chiroptical response would be well suited for the intrinsic photodetection of CPL.

We demonstrate that both chiral sidechain polymers (CSCP) and achiral polymer/chiral additive blends (ACPCA) can form chiral structures in non-aligned polymer thin films with exceptionally large chiroptical effects. We propose these effects may occur through the spatial dispersion introduced by the in-plane helical periodic modulation of polymer fibrils in a double twist cylinder-type blue phase. In contrast, when an alignment layer is introduced, the chiroptical effects in transmission occur due to the creation of a mesoscopic cholesteric-like stack normal to the substrate plane, where the twist and pitch of the polymer layers dictate the size of the chiroptical response through dissymmetric reflection of left- and right-handed CPL. These models unify the understanding of aligned and unaligned CSCP and ACPCA systems. The discovery that magneto-electric coupling—and not longer-range structural chirality—is responsible for the high dissymmetry of non-aligned chiral polymers will allow the rational design of polymers for a range of device applications. For example, the optical models described in this study can be combined with measurements of the dielectric tensor of other polymer systems to calculate their chiroptical response in reflection and transmission. Further, we anticipate that these findings will inform the design of new conjugated polymers and device architectures, where chemical structure and backbone conformation have been optimised to maximise magneto-electric coupling, allowing for strong chiroptical effects without the need for alignment and excessively thick active layers.

## Methods

**Synthesis of cPFBT and cPFO.** (S)-3,7-Dimethyloctyl bromide was synthesised based on the procedure by George, Balasubramanian and co-workers[61]. 2,7-Dibromo-9,9-bis[(3 S)-3,7-dimethyloctyl]-9H-fluorene was synthesised based on the procedure by Fukiji, Nomura and Yamada[62]. Polymers were purified at room temperature using chloroform as solvent on a preparative GPC: liquid chromatography apparatus (JAI LaboACE LC-5060 series) equipped with a pump (P-LA60, flow rate 10 ml min$^{-1}$), a UV detector (UV-VIS4ch LA, λ = 210 nm, 254 nm, 330 nm, 400 nm) and two columns (Jaigel 2HR and 2.5HR, inner diameter 20 mm × length 600 mm each). Flash chromatography was performed on Fluorochem Silica Gel 40–63 μm particle size using a forced flow of eluent at 0.3–0.5 bar

pressure[63]. Bis(1,5-cyclooctadiene)nickel(0) (CAS 1295-35-8) was purchased from Sigma and taken from a previously unopened bottle. 4,7-Dibromobenzo[c]-1,2,5-thiadiazole was provided by Cambridge Display Technology. Tris(dibenzylideneacetone)dipalladium(0) was recrystallised according to literature procedure[64]. NMR measurements were performed on a Bruker AV400 or AV500 spectrometer. Chemical shifts were referenced to the residual proton solvent peaks ($^1$H: CDCl$_3$, δ 7.26), solvent $^{13}$C signals (CDCl$_3$, δ 77.16)[65]. Signals are listed in ppm, and multiplicity identified as s = singlet, br = broad, d = doublet, t = triplet, q = quartet, quin = quintet, sep = septet, m = multiplet; coupling constants in Hz; integration. Concentration under reduced pressure was performed by rotary evaporation at around 40 °C at the appropriate pressure. Purified compounds were further dried under high vacuum (0.1–0.01 mbar). Further details on the synthesis are provided in the Supporting Information.

**Solution preparation and thin film deposition.** PFO (M$_P$ = 59 K), F8T2 (M$_W$ = 54 K), F8BT (M$_W$ = 31 K), aza[6]H were dissolved in toluene to a concentration of 20 mg/ml and blended to form a 10% (wt%) aza[6]H solution. The achiral polymers were provided by Cambridge Display Technology (CDT). cPFO and cPFBT were dissolved in toluene to a concentration of 20 mg/ml. The rubbed alignment layer was drop-cast from a dilute polyimide sealing resin based on N-methyl-2-pyrrolidone was purchased from Supelco (product number 23817). Fused silica substrates were rinsed in an ultrasonic bath in acetone and isopropyl alcohol for 20 min each, which was repeated three times. They were then transferred to an oxygen plasma asher for 5 min at 80 W before spin-coating. Thin films were spin-coated at 2,000 rpm for 60 s. Annealing took place in a nitrogen glovebox, with <0.1 ppm H$_2$O and O$_2$.

**Photophysical and morphological characterisation.** Steady-state circular dichroism (CD) measurements were performed using an Applied Photophysics Chirascan spectrophotomer. For films demonstrating strong chiroptical activity (> 2000 mdeg), a correction factor is used to calculate the true ΔA and accurate CD$_{mdeg}$[66]. In situ CD measurements were carried out at the B23 beamline at the Diamond Light Source using a vertical sample compartment for CD mapping with the XY motorised temperature controller Linkam MDS600 stage purged with N$_2$ gas. Details of the experimentally set up can be found in references[36,37]. For the in situ heating/cooling measurements, the heating rate was 10 °C/min and the samples were held at a given temperature for 1 min before spectra were acquired. For the in situ time-dependent measurements, the films were held at a given temperature and CD was acquired at a single wavelength every 1.25 s. The dissymmetry factor ($g_{abs}$) is calculated from $g_{abs}$ = ΔA/A, where A = ½ (A$_L$ + A$_R$), which refer to absorption of left- and right-handed light, respectively. Ellipticity in millidegrees can be simply calculated using $CD_{mdeg} = CD_{ΔA} \times 32982$.

Absorption measurements were acquired using an Agilent Technology Cary 300 UV–vis spectrometer. Film thicknesses were measured using a DektakXT surface profilometer.

Mueller Matrix Spectroscopic Ellipsometry was performed using a J.A. Woollam RC2 spectroscopic ellipsometer (model DI). This instrument collects 1091 data points from 193 nm to 1690 nm using a silicon CCD and InGaAs detector array for wavelengths shorter and longer than 1000 nm, respectively. Wavelength spacing is 1 nm and 2.5 nm on the Si and InGaAs detectors, respectively. Data are collected using a dual-rotating compensator ellipsometer configuration to measure all 15 normalised elements of the Mueller matrix simultaneously. All data were collected using a collimated beam with diameter of 3–4 mm. Measurements were collected at variable angles of incidence and variable azimuthal rotations for both the specularly reflected beam and transmitted beam.

Resonant Carbon K-edge soft X-ray scattering measurements were performed on the (11.0.1.2) soft x-ray scattering beamline at the Advanced Light Source of the Lawrence Berkeley National Laboratory. The x-ray energy was held at the carbon K-edge resonance (283.5 eV) and samples were drop-cast on clean silicon nitride membrane windows (Silson). Scattering intensity was collected in two-dimensions using a Princeton CCD (full experimental details provided in reference[48]). The beam is linearly polarised and can collect diffraction ring arcs in both horizontal to vertical polarisations.

AFM measurements were carried out using an Asylum Cypher S atomic force microscope (Oxford Instruments-Asylum Research, Santa Barbara, USA) operating under ambient conditions. The images (512 × 512 pixels) were acquired in AC (tapping) mode with a Scout 70 R cantilever from Nunano with a spring constant of 2 Nm$^{-1}$ and a resonant frequency of 70 kHz. The typical scan rate was 2.4 Hz. The data were processed using Gwyddion, a modular program for SPM data visualisation and analysis.

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

## Acknowledgements

J.W., J.R.B., L.W., F.S., X.S., S.T.J.R., A.J.C., and M.J.F. would like to thank the Engineering and Physical Science Research Council for funding (EP/P000525/1, EP/L016702/1, EP/R00188X/1, EP/R021503/1), as well as Cambridge Display Technology Limited (company number 02672530) for providing the achiral polymers. L.L.P., P.H.B., and D.B. A. thank the Schools of Physics and Chemistry, and the Propulsion Futures Beacon of Excellence at the UoN for supporting their research, as well as the EPSRC (EP/R513283/1). P.H.B. thanks the Leverhulme Trust for the award of a Research Fellowship [RF-2019-460]. O. A. thanks the Ministerio de Ciencia Innovación RTI2018-098410-J-I00 (MCIU/AEI/FEDER, UE). This research used resources of the Advanced Light Source, a U.S. DOE Office of Science User Facility under contract no. DE-AC02-05CH11231. We are grateful to Doug Marshall and Applied Photophysics for their support in calculating the true CD in samples showing strong dissymmetry, and Dr Chenhui Zhu for help with the RSoXS measurements. We also thank Prof Martin Heeney and Prof Iain McCulloch and their groups for helpful discussions regarding the synthesis of cPFO and cPFBT and for the use of the preparative GPC, and Dr Rohanah Hussain of Diamond B23 beamline for her assistance on beamtimes SM20376, and SM21822. We are also thankful to Thomas Penfold at the Newcastle University for his useful insight on transition dipole moments.

## Author contributions

J.W., J.R.B., A.J.C., and M.J.F. developed the concepts behind this research. J.R.B. and S.T.J.R. synthesised the chiral polymers (cPFO and cPFBT). J.W. and X.S. fabricated the samples and performed the spectroscopic experiments. J.W., J.R.B., X.S., L.W., F.S., T.J., and G.S. performed the in situ optical measurements. J.N.H. and S.S. performed the MMSE measurements and completed the data analysis in collaboration with J.W. and O.A. J.W and C.W. performed and interpreted the RSoXS measurements. L.L.-P., D.B.A., and P.H.B. performed and analysed the AFM measurements. A.J.C. and M.J.F. supervised the study and obtained funding. All authors contributed to the writing of the manuscript.

## Competing interests

A. Campbell and M. Fuchter are inventors on a patent concerning chiral blend materials (WO2014016611). The remaining authors declare no competing interests.
