## [Peer Review File · Nature Communications]

REVIEWER COMMENTS

Reviewer #1 (Remarks to the Author):

This manuscript describes a thorough study of the origin of chiroptical effects in polymers. It was found that large chiroptical effect can be observed both in polymers with chiral side chains, as well as achiral polymers to which chiral dopant is added, even in non-aligned films. The findings are very important in various applications of chiral polymers however they may also be relevant to achiral nanoparticles that are coated with chiral ligands.

The data are convincing and the analysis supports the conclusions.

This is a very important work and it should be published.

Ron Naaman

Reviewer #2 (Remarks to the Author):

The manuscript "A unified model to explain the large chiroptical effects in polymer systems through natural optical activity" by Wade et al. describes in depth studies, using the most advanced methods currently available, of optical properties of functional polymeric materials. Both the materials and methods are of timely interest. The conclusions presented, if indeed sufficiently supported by experiments, would force the community to rethink its approach to these materials and may open opportunities to fully new applications in the realm of optoelectronics.

Yet in present form I cannot recommend publication. In my view the argumentation presented by the authors for their new disruptive interpretation is still incomplete with regards to the three points described below. I would guess that the authors should be able to provide complementary data and interpretations within a reasonable amount of time and would be happy to reconsider the modified, augmented work.

1. The authors argue that magneto-electric coupling – and not longer-range structural chirality – is responsible for the high dissymmetry with respect to the absorption of left and right circular polarized light of non-aligned chiral polymer films. Fortunately, a very simple experimental test is known that can help to further discriminate between these two mechanisms. In the case of long-range structural chirality being responsible for the chiroptical properties, the dissymmetry factor should change with film thickness, tending towards zero for very thin films. In contrast, if the coupling between local electric and magnetic transition dipole moments is the active mechanism then the g_{abs} value should be largely independent of the thickness of the film. I have scanned the manuscript but cannot find information on a dedicated and concerted effort to study this thickness dependence. Yet for a fully convincing argumentation in favor of the proposed mechanism a clear graph showing dissymmetry ratio versus film thickness is mandatory.

2. On page 8 of their manuscript, the authors provide an argument against long-range helicoidal order being responsible for the circular differential response because their reflection data do not show the pronounced circular selective reflection predicted by the model.

This argument is indeed valid for polymer film consisting of

essentially of a single domain. In multidomain films with domain size smaller than the coherence length of the light, circular selective reflection can for simple geometric reasons not occur.

So, in my view, the author should clarify whether they have implicitly assumed a monodomain for their film. If so, they should also address the case of a multidomain films and provide spectroscopic

modelling appropriate for the multidomain case.

3. Finally, in earlier research on chiroptical spectroscopy, a compelling argument has been formulated which states that for molecules and materials featuring an electric dipole allowed transition, the dissymmetry factor or g-value in absorption near that dipole allowed transition must be limited and restricted to magnitudes of order 10^{-2} or lower. The argument is based on the fact that quantum mechanics limits the magnitude of the magnetic transition dipole moment to just a few Bohr magneton at most. Note also that circular polarized photons carry a limited amount of angular momentum also restricting the possibilities for realizing large magnetic transition dipole moments. The magnitude of the dissymmetry factor in the electric-magnetic mechanism depends on the relative contributions of the electric and magnetic transition dipoles to the transition probability and is largest when their contribution is about equal.

The polyfluorene polymers studied here have very strong dipole allowed transitions with large electric transition dipole moments (at least several Debye in magnitude). Therefore if indeed the electric-magnetic coupling mechanism is active it follows that the electronic transitions induced in the polymer would need to have enormous magnetic transition dipole moments that are way larger than the few Bohr magnetons indicated above .

Therefore in order for the arguments of the authors in favor of the electric-magnetic coupling mechanism to be viable, the authors need to provide an account of the magnitude of the transition dipole moments required and how such magnitude may be realized in the π -conjugated polymers under study.

Stefan Meskers

Reviewer #3 (Remarks to the Author):

This paper quantitatively accounts for very large optical activity of a set of polymeric films, which have been demonstrated (or can be) interesting for highly innovative applications as active layers of organic electro-optical devices.

The key question is if the observed optical activity stems from the "natural" coupling of magnetic/electric dipoles of the Rosenfeld equation or rather from the twisted dielectric, as it occurs in a chiral nematic.

The fact may have profound consequences in the rational quest for new materials for chiral organic electro-optical devices. Moreover, natural optical activity corresponds to a real process of CP-selective photon absorption or emission, whereas structural chirality effects are rather based on dispersion/reflection processes (i.e. photon losses)

Notably, the authors observe that the thickness of a film for practical uses in electronic devices should be <200 nm, which is much shorter than the wavelength of light and is less compatible with efficient Bragg reflection.

Indeed, the twisted dielectric mechanism should display a marked dependence of the g-anisotropy factor on film thickness, unlike natural optical activity: this would provide a very simple expedient for discriminating between the two processes and I should warmly invite the authors to provide evidence in this sense.

The authors study two classes of systems: achiral polymers with a chiral additive and polymers, where the chiral element is covalently embedded. Not very surprisingly, they find no major difference between the two cases: in my modest opinion, as long as the chiral additive is well-dissolved and well-solvated by the polymer, covalency or not makes a lesser role to me. But let's leave my personal expectations aside.

The authors take fundamental advantage of Mueller matrix polarimetry both in absorption and in reflection and demonstrate that, while in absorption one can more or less fit the observed curves with

both models (natural optical activity vs. structural chirality), in reflection structural chirality fails. This is the crucial answer sought.

When an alignment layer is used to induce preferential orientation, things become different, because linear anisotropies very obviously arise but, and this is more relevant, the structural chirality (twisted dielectric model) takes a much more significant role.

The manuscript contains a long series of complementary characterization, but I skip their analysis or comment for the sake of brevity.

I may not fully appreciate how this is a "unified model", as it seems to me rather based on two parallel and possibly concurring/competing mechanisms. Significantly, the word unified (or similar) only occurs in the title.

This contribution is interesting to me for the care, completeness and sophistication of the experiments and of the reasonings and I should welcome its publication in Nature Communications, after considering the suggestions below.

I would also comment that it is not always easy to follow the text for the shift between main text and supplementary information, whereby much of the latter one is indeed essential.

- I should recommend reporting the CD measurements as a function of film thickness for both the various sets of samples.
- Fig. S2 should report the wavelength of the CD data reported in this spatially-resolved CD data.
- Throughout the paper, the authors discuss the " π - π^* " transition, as if it were unique. Most of the transitions in these systems are indeed π - π^* . It should rather be referred to as the lowest energy, most redshifted ...

Response to reviewers:

Reviewer #1 (Remarks to the Author):

This manuscript describes a thorough study of the origin of chiroptical effects in polymers. It was found that large chiroptical effect can be observed both in polymers with chiral side chains, as well as achiral polymers to which chiral dopant is added, even in non-aligned films. The findings are very important in various applications of chiral polymers however they may also be relevant to achiral nanoparticles that are coated with chiral ligands.

The data are convincing and the analysis supports the conclusions.

This is a very important work and it should be published.

Ron Naaman

We are very grateful for your kind remarks.

Reviewer #2 (Remarks to the Author):

The manuscript "A unified model to explain the large chiroptical effects in polymer systems through natural optical activity" by Wade et al. describes in depth studies, using the most advanced methods currently available, of optical properties of functional polymeric materials. Both the materials and methods are of timely interest. The conclusions presented, if indeed sufficiently supported by experiments, would force the community to rethink its approach to these materials and may open opportunities to fully new applications in the realm of optoelectronics.

Yet in present form I cannot recommend publication. In my view the argumentation presented by the authors for their new disruptive interpretation is still incomplete with regards to the three points described below. I would guess that the authors should be able to provide complementary data and interpretations within a reasonable amount of time and would be happy to reconsider the modified, augmented work.

We are grateful for the suggested changes and agree that they will greatly improve the manuscript.

1. The authors argue that magneto-electric coupling – and not longer-range structural chirality – is responsible for the high dissymmetry with respect to the absorption of left and right circular polarized light of non-aligned chiral polymer films. Fortunately, a very simple experimental test is known that can help to further discriminate between these two mechanisms. In the case of long-range structural chirality being responsible for the chiroptical properties, the dissymmetry factor should change with film thickness, tending towards zero for very thin films. In contrast, if the coupling between local electric and magnetic transition dipole moments is the active mechanism then the g_{abs} value should be largely independent of the thickness of the film. I have scanned the manuscript but cannot find information on a dedicated and concerted effort to study this thickness dependence. **Yet for a fully convincing argumentation in favor of the proposed mechanism a clear graph showing dissymmetry ratio versus film thickness is mandatory.**

We thank the reviewer for highlighting that we did not adequately discuss the thickness-dependence of the chiroptical response. We had previously reported the thickness-dependence on g_{abs} (10.1021/acsnano.9b02940) for F8BT-based ACPCA systems but have decided to include a more thorough analysis in the new version of the manuscript based on your comments. The new analysis follows the reflectance correction protocol proposed by Schiek *et al* (10.1038/s41467-018-04811-7), to accommodate for reflection losses at the interfaces. This correction protocol involves calculating reflection-corrected absorbance ($\text{Abs}_{\text{corrected}}$) by obtaining the gradient of peak absorbance vs. film thickness and using this $\text{Abs}_{\text{corrected}}$ to calculate g_{abs} . Further details of the protocol followed can be found here (10.1038/s41467-018-04811-7).

To investigate the impact of film thickness on the strength of the chiroptical response we created a series of films with different thicknesses and measured the circular dichroism. We calculated the associated g -factors of films annealed at $T_{\text{CD Max}}$ by normalising the CD (ΔA) to the unpolarised absorbance of the

sample. These uncorrected g -factors (both for aligned and not-aligned systems) appear to increase with increasing film thickness. However, once reflection losses are accounted for, the **non-aligned** thin films show constant g_{abs} as a function of film thickness, whereas **aligned** thin films show a linearly increasing g_{abs} . We should note that for the aligned (i.e. structurally chiral) case, the dissymmetry tends towards but does not reach zero (intercept of aligned F8BT:aza[P] ~ 0.3 , cPFBT ~ 0.6) for very thin films. This could be due to a combination of structural chirality and natural optical activity influencing the chiroptical response, or changes in thin film morphology as a function of film thickness – particularly for thin films deposited on top of alignment layers. We should note that it was not possible to measure the thickness-dependent dissymmetry of aligned PFO/cPFO films due to overlap between the alignment layer (polyimide) and polymer layer absorption.

The following graphs have been included in the **Supporting Information** with the accompanying text.

*“To quantitatively assess the optical activity of aligned and non-aligned ACPA and CSCP thin films annealed at $T_{CD Max}$ we calculated the dissymmetry factor, g_{abs} , by normalising the CD (ΔA) to the unpolarised absorbance of the sample. To better understand the origins of this chiroptical response, we controlled the spin-coating speed to fabricate a series of thin films with different thicknesses. At first glance, the g -factors of all systems (aligned and non-aligned, ACPA and CSCP) appear to increase as a function of film thickness (i.e. thicker films achieve higher g_{abs} than thin films). To accommodate for reflection losses at the interfaces, we followed the simple protocol introduced by Schiek et al. Once reflection losses are accounted for, the **non-aligned** thin films have a thickness-independent g_{abs} , whereas **aligned** thin films show a g_{abs} that increases linearly as a function of film thickness. Clearly, this thickness-dependent dissymmetry of aligned systems (i.e. where structural chirality dominates) should tend to zero for very thin*

films, but in the cases considered here, we believe both natural optical activity and structural chirality contribute to the measured chiroptical response.”

The following is now in the **main manuscript**,

*“To further confirm that it is natural optical activity not structural chirality that gives rise to the strong chiroptical response of non-aligned chiral polymer films, we measured the g-factors of aligned and non-aligned thin films of different thicknesses (see Figure S25 and accompanying discussion). Once reflection losses are accounted for, the **non-aligned** thin films have a thickness-independent g_{abs} , as expected from natural optical activity, whereas **aligned** thin films show a g_{abs} that increases linearly as a function of film thickness.”*

2. On page 8 of their manuscript, the authors provide an argument against long-range helicoidal order being responsible for the circular differential response because their reflection data do not show the pronounced circular selective reflection predicted by the model. This argument is indeed valid for polymer film consisting of essentially of a single domain. In multidomain films with domain size smaller than the coherence length of the light, circular selective reflection can for simple geometric reasons not occur. **So, in my view, the author should clarify whether they have implicitly assumed a monodomain for their film. If so, they should also address the case of a multidomain films and provide spectroscopic modelling appropriate for the multidomain case.**

Thank you for highlighting this shortcoming in our manuscript, indeed, we are aware of the extensive study into multi-domain ‘mosaic’ cholesteric liquid crystal thin films, in particular the recent work of Di Nuzzo *et al* (10.1021/acsnano.7b07390). In general, our film thicknesses (T) are < 150 nm, i.e. they are considerably thinner than those considered in the Di Nuzzo model. As described in our paper, we see no evidence of linear dichroism; not only in rotating or flipping the samples, but also in spatially resolved measurements at the Diamond Light Source (Figure S2), where the beam diameter was 0.01–0.05 mm. Further, RSoXS measurements indicate no long-range helical order perpendicular to the substrate plane, spatially resolved CD measurements indicate a homogenous conformation and the optical texture seen in the optical micrographs is in favour of a blue-type phase rather than cholesteric. Taken together, our results indicate that neither a mono- or multi-domain cholesteric stack is present in the non-aligned ACPA or CSCP systems.

At the request of the reviewer, a helically multi-domain model was created that allowed mixing of multiple domains with ability to vary their 1) starting orientation, 2) amount of twisting, and 3) tilt angle. We found that this multi-domain model cannot simultaneously justify the absence of circular differential effects in reflection, without also eliminating the transmission circular effects. Furthermore, given the short coherence length of the ellipsometer source and the large beam size, it is expected that an eventual multi-domain situation could easily lead to a depolarizing optical response if there was some degree of incoherence between the contributions of the different domains, as it was first described by Dmitrienko and Belyakov in 1977.

“a qualitative new phenomenon not observed in perfect CLC is the depolarization of light in a mosaic crystal. It must be noted that in a mosaic CLC, besides the usual beam depolarization due to the inhomogeneities, a depolarization is caused by diffractive scattering,”

This is **not observed** in our data, as evidenced by the following figures that consider an incoherent multidomain model.

Details of this multi-domain model, including a comparison of simulated (mono- and multi-domain) and experimental spectra, are now provided in the **Supporting Information (Figure S8)**:

To establish whether a multi-domain or ‘mosaic’ cholesteric liquid crystal structure could be responsible for the strong chiroptical phenomena recorded here, a model was created that allowed mixing of multiple domains with ability to vary their 1) starting orientation, 2) amount of twisting, and 3) tilt angle. Of these three effects, calculations with multiple “starting orientations” can effectively minimize the linear response for transmitted measurements, while maintaining the circular response.

- **Graded Layer** Thickness # 1 = 131.42 nm
 - Grade Type = Parametric # of Slices = 51
 - Profile = Custom
 - Material = Biaxial
 - Type = Uniaxial
 - Optical Constants: Difference Mode = OFF
 - + Ex = F8BT_p10_XY_Genosc
 - + Ez = F8BT_p10_Z_Genosc
 - Euler Angles: Phi = (Graded) Theta = 90.000 (fit)
 - Grading Parameters:** Add Delete Delete All

Name	Top Value	Graph
Phi	[bottom]+[pos]*([turns]*360)	Draw

 - Custom Grade Fit Parameters**
bottom = 0.0000 (fit) turns = 0.42000 (fit)
- + Substrate = EMA

- **MODEL Options**

Angle Offset = 0.00
 Include Substrate Backside Correction = OFF
 Model Calculation = Ideal

- **Parameter Smearing**

Parameter to Smear: bottom # Values = 30 Smear Width = 360.000
 2nd Parameter to Smear: Theta # Values = 10 Smear Width #2 = 30.000
 3rd Parameter to Smear: turns # Values = 20 Smear Width #3 = 0.500

Details of the multi-domain model that modifies for starting orientation.

MMSE spectra recorded in transmission and a model that makes use of a (top) single- and (bottom) multi-domain cholesteric stack with 15 different starting angles.

Measured **transmission** depolarisation index (pink line) and the predicted depolarisation index for single (black line) and multi-domain (blue line) systems.

Whilst a multi-domain model that only makes use of multiple starting orientations can minimise the linear terms of transmission measurements, it does not suppress the circular response of the reflection spectra. Instead, we found a combination of all three (i.e. a calculation that allows for starting orientation, amount of twist, and tilt angle) **can** suppress the circular response. However, it would suppress the circular response for both transmitted and reflected beams and would lead to significant depolarization.

- Graded Layer Thickness # 1 = 141.86 nm
 - Grade Type = Parametric # of Slices = 51
 - Profile = Custom
- Material = Biaxial
 - Type = Uniaxial
 - Optical Constants: Difference Mode = OFF
 - + Ex = F8BT_p10_XY_Genosc
 - + Ez = F8BT_p10_Z_Genosc
 - Euler Angles: Phi = (Graded) Theta = 90.000 (fit)
- Grading Parameters: Add Delete Delete All

Name	Equation	Graph
Phi	[bottom]+[pos]*(360*[twists])	Draw

- Custom Grade Fit Parameters
 - bottom = 18.56920 (fit) twists = 0.30000 (fit)
- + Substrate = EMA

- **MODEL Options**
 - Angle Offset = 0.00
 - Include Substrate Backside Correction = OFF
 - Model Calculation = Ideal
- **Parameter Smearing**
 - Parameter to Smear: bottom # Values = 15 Smear Width = 180.000
 - 2nd Parameter to Smear: twists # Values = 10 Smear Width #2 = 2.000
 - 3rd Parameter to Smear: Theta # Values = 10 Smear Width #3 = 45.000

Figure 1: **Reflection** M14 (CD) and M23 (CB) terms predicted using the multi-domain model.

Figure 2: Measured **reflection depolarisation index** (pink and blue lines) and the modelled depolarisation index for a single (left) and multi-domain (right) cholesteric stack.

Figure 3: **Transmission MMSE** spectra recorded for annealed F8BT:aza[M] films and the predicted spectra from a model that controls for the 1) starting orientation, 2) amount of twist, and 3) tilt angle.

In summary, we have not been able to find a model based on a multi-domain system that simultaneously satisfies all the experimental observations: 1) suppress the **linear anisotropic responses** that would be obtained in transmission MMSE measurements, 2) suppress the **circular anisotropic responses** that would be obtained in reflection MMSE measurement, 3) maintain the chiroptical response in transmission (not only at normal incidence but also at oblique incidence) and 4) not introducing **significant depolarization**.

The following text has been included in the **Main Manuscript**.

“We also attempted to model the data using a helical multi-domain model (Figure S8 and accompanying discussion) but were not able to satisfy all of the experimental results without introducing significant depolarisation, which is not observed in our measurements.”

3. Finally, in earlier research on chiroptical spectroscopy, a compelling argument has been formulated which states that for molecules and materials featuring an electric dipole allowed transition, the dissymmetry factor or g-value in absorption near that dipole allowed transition must be limited and restricted to magnitudes of order 10^{-2} or lower. The argument is based on the fact that quantum mechanics limits the magnitude of the magnetic transition dipole moment to just a few Bohr magneton at most. Note also that circular polarized photons carry a limited amount of angular momentum also restricting the possibilities for realizing large magnetic transition dipole moments. The magnitude of the dissymmetry factor in the electric-magnetic mechanism depends on the relative contributions of the electric and magnetic transition dipoles to the transition probability and is largest when their contribution is about equal. The polyfluorene polymers studied here have very strong dipole allowed transitions with large electric transition dipole moments (at least several Debye in magnitude). Therefore if indeed the electric-magnetic coupling mechanism is active it follows that the electronic transitions induced in the polymer would need to have enormous magnetic transition dipole moments that are way larger than the few Bohr magnetons indicated above .

Therefore in order for the arguments of the authors in favor of the electric-magnetic coupling mechanism to be viable, the authors need to provide an account of the magnitude of the transition dipole moments required and how such magnitude may be realized in the π -conjugated polymers under study.

We agree that attributing the origins of the strong chiroptical responses of these transitions to natural optical activity is in apparent contrast to the quantum mechanical description described by the reviewer above. In general, though, aggregated chromophores that adopt a chiral arrangement can show very large natural optical activity. It is plausible, given the highly anisotropic local environment in the condensed phase, that the chromophore may possess no local symmetry, in which case, transitions are magnetically and electrically allowed. We believe that the chiroptical response of these ‘inherently dissymmetric chromophores’ (10.1016/B0-12-369397-7/00084-4) is enhanced *via* exciton coupling (10.1002/9781118120392.ch4, 10.1039/B515476F) in the double twist cylinder blue phase. A more detailed investigation of the precise nature of the electro-magnetic coupling is ongoing and believe it to lie beyond the scope of this communication.

Nonetheless, to address this concern, we have included the following text in the **Main Manuscript**.

“The very large chiroptical effects associated with natural optical activity that we report contrasts with the small effects typically observed for small molecules in solution, but is reminiscent of the high optical activity of chiral aggregates where supramolecular chirality leads to CD effects that are magnitudes higher than those of the isolated molecules.²⁻⁵ For unaligned polymer films, the in-plane helical periodic modulation of the polymer in the double twist cylinder blue phase may give rise to a chromophore large enough to possess no local symmetry, meaning that transitions are electrically and magnetically allowed.⁶ Exciton coupling of nearby polymer chains, as evidenced by the positive and negative Cotton bands in the CD spectra (Figure 2), may serve to further enhance the chiroptical response.”

*The following references have been added to the **Main Manuscript**:*

1. Dmitrienko, V. D. & Belyakov, V. A. Contribution to the theory of the optical properties of imperfect cholesteric liquid crystals. *Sov. Phys. JETP* **46**, 362–356 (1977).
2. Liu, M., Zhang, L. & Wang, T. Supramolecular chirality in self-Assembled systems. *Chemical Reviews* **115**, 7304–7397 (2015).
3. Veling, N. *et al.* Solvent-dependent amplification of chirality in assemblies of porphyrin trimers based on benzene tricarboxamide. *Chem. Commun.* **48**, 4371 (2012).
4. Danila, I. *et al.* Hierarchical Chiral Expression from the Nano- to Mesoscale in Synthetic Supramolecular Helical Fibers of a Nonamphiphilic C₃-Symmetrical π -Functional Molecule. *J. Am. Chem. Soc.* **133**, 8344–8353 (2011).
5. Gottarelli, G., Lena, S., Masiero, S., Pieraccini, S. & Spada, G. P. The use of circular dichroism

spectroscopy for studying the chiral molecular self-assembly: An overview. *Chirality* **20**, 471–485 (2008).

6. Kuball, H.-G. CHIROPTICAL ANALYSIS. in *Encyclopedia of Analytical Science* (ed. Townsend, A.) 60–79 (Elsevier, 2005).

Reviewer #3 (Remarks to the Author):

This paper quantitatively accounts for very large optical activity of a set of polymeric films, which have been demonstrated (or can be) interesting for highly innovative applications as active layers of organic electro-optical devices.

The key question is if the observed optical activity stems from the “natural” coupling of magnetic/electric dipoles of the Rosenfeld equation or rather from the twisted dielectric, as it occurs in a chiral nematic. The fact may have profound consequences in the rational quest for new materials for chiral organic electro-optical devices. Moreover, natural optical activity corresponds to a real process of CP-selective photon absorption or emission, whereas structural chirality effects are rather based on dispersion/reflection processes (i.e. photon losses)

Notably, the authors observe that the thickness of a film for practical uses in electronic devices should be <200 nm, which is much shorter than the wavelength of light and is less compatible with efficient Bragg reflection.

Indeed, the twisted dielectric mechanism should display a marked dependence of the g-anisotropy factor on film thickness, unlike natural optical activity: this would provide a very simple expedient for discriminating between the two processes and I should warmly invite the authors to provide evidence in this sense.

The authors study two classes of systems: achiral polymers with a chiral additive and polymers, where the chiral element is covalently embedded. Not very surprisingly, they find no major difference between the two cases: in my modest opinion, as long as the chiral additive is well-dissolved and well-solvated by the polymer, covalency or not makes a lesser role to me. But let's leave my personal expectations aside. The authors take fundamental advantage of Mueller matrix polarimetry both in absorption and in reflection and demonstrate that, while in absorption one can more or less fit the observed curves with both models (natural optical activity vs. structural chirality), in reflection structural chirality fails. This is the crucial answer sought.

When an alignment layer is used to induce preferential orientation, things become different, because linear anisotropies very obviously arise but, and this is more relevant, the structural chirality (twisted dielectric model) takes a much more significant role.

The manuscript contains a long series of complementary characterization, but I skip their analysis or comment for the sake of brevity.

I may not fully appreciate how this is a “unified model”, as it seems to me rather based on two parallel and possibly concurring/competing mechanisms. Significantly, ***the word unified (or similar) only occurs in the title.***

Thank you for your efforts with this review, we are grateful for your comments. The phrase “unified model” was chosen to describe the identical origins of the strong chiroptical response in non-aligned CSCP and ACPCA systems, i.e. in conditions relevant for device fabrication. Indeed, the aligned scenario, where structural chirality is responsible for the strong CD, is a totally different mechanism.

We have updated the **main manuscript** to include the following statement:

“In contrast, when an alignment layer is introduced, the chiroptical effects in transmission occur due to the creation of a mesoscopic cholesteric-like stack normal to the substrate plane, where the twist and pitch of the polymer layers dictates the size of the chiroptical response through dissymmetric reflection of left- and right-handed CPL. **These models unify the understanding of aligned and unaligned CSCP and ACPCA systems.** The discovery that magneto-electric coupling – and not longer-range structural chirality –

is responsible for the high dissymmetry of non-aligned chiral polymers will allow the rational design of polymers for a range of device applications.”

This contribution is interesting to me for the care, completeness and sophistication of the experiments and of the reasonings and I should welcome its publication in Nature Communications, after considering the suggestions below.

I would also comment that it is not always easy to follow the text for the shift between main text and supplementary information, whereby much of the latter one is indeed essential.

We are grateful for your careful reading and comments; and agree, we wish we had more space in the main text to include these important observations and findings, but are bound by the requirements of Nature Communications. It is possible we may be able to address this concern through further discussions with the editor if our paper is accepted.

• I should recommend reporting the CD measurements as a function of film thickness for both the various sets of samples.

We agree that thickness dependent studies are essential to further clarify the origins of these strong chiroptical effects. Following the advice of yourself and Reviewer 2 (see response above), we have included thickness-dependent measurements of uncorrected and reflection-corrected g_{abs} in the Supporting Information. As can be seen, non-aligned samples show a thickness-independent g_{abs} , whereas aligned samples (i.e. situations where structural chirality dominates the chiroptical response) show a linearly increasing g_{abs} as a function of thickness.

• Fig. S2 should report the wavelength of the CD data reported in this spatially-resolved CD data.

Thank you very much for pointing out that we did not include the wavelength in Figure S2, this is obviously essential for the interpretation of these data. The wavelength is $\lambda = 481$ nm and the spectra were all recorded at room temperature. We have updated the **Supporting Information** as follows;

“The chiroptical response was recorded at $\lambda = 481$ nm. Homogeneity of the chiral response across these length scales sample permits further characterisation using MMSE, as it confirms that the sample is uniform within the beam diameter of the ellipsometer.”

• Throughout the paper, the authors discuss the “pi-pi*” transition, as if it were unique. Most of the transitions in these systems are indeed pi-pi*. It should rather be referred to as the lowest energy, most redshifted ...

We are grateful for your comments and agree that using the term pi-pi* can confuse the reader, especially when trying to interpret figures and graphs. As such, we have removed the phrase “ $\pi \rightarrow \pi^*$ ” from the manuscript and replaced it with **“lowest energy transition”**.

We also made minor formatting changes to the **Supporting Information**, including;

- **page 14:** formatting g -factors so that they were consistent with the main manuscript (i.e. g_{abs} not g_{abs}).
- **page 15:** removing row of Table S2 that gave the Raman shift (cm^{-1}) of the C=C mode of the conjugated polymers, as this was not used in the study.
- **page 18:** font size was changed to 12 to be consistent with the rest of the SI. Equations were aligned centre.
- **page 66:** (optical micrographs) “F8BT doped with aza[6]H (right)” was changed to “F8BT (left) and F8BT with 10 wt% aza[6]H (right)”

In the **Main Manuscript**, all the figure numbers and references were updated to reflect the changes described above.

REVIEWER COMMENTS

Reviewer #2 (Remarks to the Author):

With their revised version, Wade and coworkers have responded to the comments of the reviewers. With respect to the three points I raised in my earlier report:

1) The authors have appropriately addressed the issue of film thickness and its influence on the optical properties. Well done.

2) In the revised version the authors have now also added some modeling of multidomain films. Yet, I cannot find a reference to a detailed description of the particular models used and so I have now idea which approximations are involved. I would not be able to reproduce the modelling; the model(s) is(are) not traceable.

From the new figure S8 on page 39, it is clear that the modelling involves 'parameter smearing' which suggests that the model involves the effective medium approximation where the final result is an average over the predictions obtained by varying parameters in a certain range. This would correspond to domains of essentially macroscopic size.

In the description now provided, there is no mention of the typical size of the domains. For many optical effects, the optical properties rapidly change when the domain size crosses the (sub)micrometer range where it is comparable to the wavelength of light. Has this important range been covered in the multidomain modelling?

In summary, the authors need to make their modelling efforts traceable and convince the reader that the important case of domains with (sub)micrometer length scale has been covered.

3) The authors have, in my view, not really addressed the issue of the magnitude of the electric and magnetic dipole moments required in their proposed explanation. They do not come up with estimates for the electric and magnetic dipole moments that lie at the core of their new explanation.

The argument provided by the authors that in supramolecular chiral systems also high g-values are observed is invalid because in the supramolecular structures there may be contributions to the circular dichroism from the long-range structural chirality that the authors are trying to disprove.

In short, I am not convinced that the magnitudes of the dipole moments required in the new explanation are physically viable.

Stefan Meskers

Reviewer #3 (Remarks to the Author):

The manuscript has greatly improved, thanks to the introduction of carefully conducted and thoughtfully presented thickness-dependent experiments. In the rebuttal, very correctly, the authors put forward a relevant point: "changes in thin film morphology as a function of film thickness", which may be even more true on account of different spinning rates. Indeed, in general, competing aggregation pathways may lead to polymorphs whose relative weight may be thickness- or spinning rate-dependent. They can be expected to be associated to different CD spectra and the existence of a similar multiplicity of species can be demonstrated by studying CD profiles as a function of the thickness, of the spinning rate, of the distance from the spinning focus. All this would go beyond the present investigation but I would suggest, as a minor correction to account for this in the text. The point raised by referee 2 is correct in the absence of exciton coupling and indeed even the most extreme cases of intrinsically chiral chromophores (including helicenes and other highly conjugated twisted systems) hardly go over $g=10^2$. In some cases, chiral supramolecular architectures can overcome this limit, thanks to exciton coupling and I consider myself satisfied with the text introduced by the authors.

On account of the value of the work and of the results I'll not insist, recommending further changes, but to me a "unified model" is one, which accounts for several effects in different conditions, which is the opposite of: "Indeed, the aligned scenario, where structural chirality is responsible for the strong CD, is a *totally different mechanism*", or of "***In contrast*** when an alignment layer is introduced, the chiroptical effects in transmission occur due to the creation of a mesoscopic cholesteric-like stack normal to the substrate plane, where the twist and pitch of the polymer layers dictates the size of the chiroptical response through dissymmetric reflection of left- and right-handed CPL."

Reviewer #2 (Remarks to the Author):

With their revised version, Wade and coworkers have responded to the comments of the reviewers. With respect to the three points I raised in my earlier report:

1) The authors have appropriately addressed the issue of film thickness and its influence on the optical properties. Well done.

2) In the revised version the authors have now also added some modeling of multidomain films. Yet, I cannot find a reference to a detailed description of the particular models used and so I have now idea which approximations are involved. I would not be able to reproduce the modelling; the model(s) is(are) not traceable.

We remain grateful for the important suggestion at the first revision to include a multi-domain model. We apologise if our descriptions of the model lacked sufficient detail. We have now updated the Supporting Information to include more details about the modeling effort that was put forth in our previous response – where the linear effects are “smeared” by considering multiple grains with different properties. The mixing of multiple Mueller matrices with different values inherently leads to considerable depolarization (see below), which would have been measured from a “mosaic” multi-domain structure. Crucially, we did not find evidence of depolarization experimentally, as explained in our prior response. We have now included the following in the Supporting Information:

The multi-domain model uses an incoherent summation of calculated Mueller matrices (M_k) corresponding to individual systems with varied properties. For example, smearing of the bottom orientation was demonstrated by calculating 30 different Mueller matrices, each with a different starting molecular orientation (Phi Euler angle $\phi(n)$), with a total range of 360° . These 30 different Mueller matrices are combined incoherently. Our model also calculated the variation of two additional model parameters– the tilt axis of the liquid crystal and the total number of twists. The detected Mueller matrix (M_{det}) is a weighted summation of the individual Mueller matrices, as:

$$M_{det} = \sum_k i_k \ddot{M}_k \quad \text{Equation 1}$$
$$\sum_k i_k = 100\%$$

The mixing of multiple Mueller matrices with different values inherently leads to considerable depolarization, which would have been measured from a “mosaic” multi-domain structure with domain sizes larger than the coherence length of the light source (~ domains larger than the wavelength).

From the new figure S8 on page 39, it is clear that the modelling involves 'parameter smearing' which suggests that the model involves the effective medium approximation where the final result is an average over the predictions obtained by varying parameters in a certain range. This would correspond to domains of essentially macroscopic size.

In the description now provided, there is no mention of the typical size of the domains. For many optical effects, the optical properties rapidly change when the domain size crosses the (sub)micrometer range where it is comparable to the wavelength of light. Has this important range been covered in the multidomain modelling?

In summary, the authors need to make their modelling efforts traceable and convince the reader that the important case of domains with (sub)micrometer length scale has been covered.

We very much appreciate the further input of the reviewer on this point. We interpret their comment to suggest that a collection of sub-micron, partially formed cholesteric domains (i.e. not perfect Bragg reflectors), could smear the circular reflectance whilst maintaining the circular transmittance. Sub-micron sized domains will not generate depolarization as for larger domains, since their individual optical responses will combine coherently. At the request of the reviewer, we explored the ramifications of this model below.

We have created an optical model that can describe the optical response of a coherent superposition of multiple grains of twisted systems, where the helical pitch is not fully developed (i.e. cholesteric stacks of heights much smaller than a full pitch), and with random orientations (see Figure 1). We find that the modelled optical response of such a system *can* lead to the vanishing of circular effects in reflection, whilst maintaining the circular effects in transmission. However, it was important to apply constraints on the starting orientations for the various twisted domains to essentially eliminate the reflected response. For example, two twisted grains with starting orientation rotated 90° from each other, or three twisted grains oriented 60° apart from each other (see further discussion below). We should emphasise that we do not see the same cancellation of the circular terms in reflection where the twist is more developed (i.e. a Bragg reflector), which maintain the same optical response regardless of the in-plane orientation of the sample.

Figure 1: a cartoon of the sub-micron multi-domain cholesteric system considered in this model, which involves a coherent superposition of randomly oriented grains.

The model makes use of the Woollam WVASE® General Multi-model Patterning Layer, which allows either coherent or incoherent mixing of multiple models (which refers to the individual grains). The coherent mixing of multiple grains is described by the weighted sum of multiple (up to 5) Jones matrices:

$$J = \begin{pmatrix} j_{11} & j_{12} \\ j_{21} & j_{22} \end{pmatrix} = \sum_n c_n \begin{bmatrix} r_{pp_n} & r_{sp_n} \\ r_{ps_n} & r_{ss_n} \end{bmatrix} \quad \text{Equation 2}$$

where c_n is a weighting factor associated with each individual grain (n^{th} Jones matrix) and r_{pp} , r_{sp} , r_{ps} , and r_{ss} represent the complex Fresnel coefficients for the n^{th} model. Figure 2 describes the general approach that underpin this analysis; the coherent mixing of 4 grains – where each model represents a grain with a twisted anisotropic structure and a different starting orientation. For each grain, the total amount of twist can vary along with film thickness and anisotropic optical constants to fit the MM data in reflection (25°) and transmission (0°). Anisotropic optical constants (Figure 3) were allowed to vary in both directions but constrained by Kramers-Kronig consistent oscillator summations.

Figure 2: The multi-domain coherent model described above. The specific details of the layers involved in the model are provided below.

4	Graded (biaxial)	111.304 nm
3	biaxial	0.000 nm
2	GenOsc_Z_p10	0.000 nm
1	GenOsc_XY_p10	0.000 nm
0	fused silica	1.5 mm

Figure 3: Details of the optical constants used for the multi-domain model.

Figure 4 shows the MMSE data collected in transmission (angle of incidence 0°) and reflection (angle of incidence 25°) for a F8BT:aza[M] film alongside the coherent multi-domain cholesteric model proposed above. We then focus on the **transmitted** and **reflected** circular responses (M_{14} and M_{23}) for a single model (grain) and the situation when four grains are combined coherently (Figure 5). Finally, we consider the circular response (M_{14} and M_{23}) in reflection for a single model (grain) using the same total twist (105°) but with different starting orientations (Figure 6). If the data averaged more grains of one orientation than another, then cancellation would not be complete, and we would see a preferential circular response in reflection.

Figure 4: Simulated (using the multi-domain model) and experimental MMSE spectra in transmission and reflection (25°).

Transmitted circular response (M_{14} and M_{23})

Reflected circular response (M_{14} and M_{23})

Figure 5: transmitted and reflected circular response for individual ('model 3') and coherently combined ('model 3 + 6') grains.

Figure 6: The circular response (M_{14} and M_{23}) in reflection for a single grain.

Based on this data, we can draw the following conclusion. For the cancellation to be effective, the twisted structures need to “balance” each other out with complementary starting orientations. For example, a starting orientation of 0° can be balanced with a starting orientation of 90° . It is very unlikely that there are a small number of starting orientations – especially since the in-plane anisotropy of this twisted model also produces linear optical responses which need to be smeared out (in addition to the circular responses). It is more likely that the coherent models would be a collection of random orientations and possibly random tilt angles relative to the surface.

Informed by the coherent multi-domain model described above, we compared the simulated and experimental transmission data at variable angles (transmission, MMr; 0° , 30° and 60° and reflection, MMr; 25° and 60° , Figure 7). If multiple angles of incidence are considered; the coherent multi-domain model fails to fit all data

together. This indicates that this multidomain model is not compatible with the overall uniaxial response seen on the sample.

Transmission (0, 30, 60°)

Reflection (25, 60°)

Figure 7: Experimental and simulated transmission and reflection MMSE spectra acquired at multiple angles of incidence

In order to further experimentally investigate whether the optical response from such a multi-domain structure exists within our samples, we mapped the uniformity of the circular terms in reflectance (65°) using a 40 μm beam over 1200 points on a 0.8 mm × 0.8 mm area. Whilst this is by no means a sub-micron measurement, if the grain sizes are in the micron-range then such a focused measurement should show varying circular

effects when not perfectly “cancelled”. The grey circles (the vertical line along $x = 0$) were selected to compare spectra.

As can be seen below in Figure 8, there is no indication of large grains such as those in Figure 6 (the circular response of single grains) where the circular response would not maintain “balance” at different locations. In other words, any proposed multi-grain model would need to contain a large number of smaller grains to smear-out the circular reflected response in a similar manner at each of these points.

Figure 8: Spatially resolved circular terms measured in reflectance for an F8BT:aza[M] thin film.

Comparable to the spatially resolved CD spectra acquired at the Diamond Light Source (Figure S2) in the Supporting Information, the response is incredibly uniform. In fact, the new, high resolution, spatially resolved spectroscopic data (Figure 8) perfectly fits the model included in our original submission (Figure 9), which incorporates magneto-electric coupling (i.e., the natural optical activity described in the paper).

Figure 9: Simulated MMSE spectra in reflection (65°) using the 6×6 magneto-electric coupling

Finally, we have also measured transmission and reflection MMSE data for ultra-thick films (915 nm), which should be closer to a ‘Bragg reflector’ (Figure 10). Even at these thicknesses, there is no circular selective response in reflection.

Transmission (0°)

Reflection (15°)

Figure 10 Transmission and reflection MMSE spectra recorded for a 915 nm thick F8BT:aza[6]H (P) film

We are immensely grateful to the reviewer for their suggestion to pursue the modelling further. Details of the partially-cholesteric multi-domain model with random starting orientations are now provided in the SI. In addition to the inconsistencies between the new multi-grain model and the experimental MMSE data, neither AFM, cross-polarised or spatially resolved CD indicate ordering on these length scales. Instead these are consistent with the formation of a double-twist cylinder blue phase (Figure 6 of main manuscript), which is supported by an optical model that involves magneto-electric coupling.

The following text has been added to the Supporting Information, along with details of the multi-domain model described above.

Alongside investigating the optical response of a single domain cholesteric model, two multi-domain models were generated, and the Mueller Matrix evaluated as an (1) an incoherent (domains > the wavelength of light) and (2) a coherent (domains < the wavelength of light) superposition. We should note, at present there are no experimental control samples to validate the outputs of either model. We find that the coherent superposition (2) of randomly oriented, sub-micron, partially formed cholesteric domains (i.e. not fully complete twists) can result in MMSE spectra without circular elements, but only at certain angles of incidence. The model does not hold when multiple angles of incidence are considered and we see no evidence of such grains through AFM, cross-polarised microscopy or spatially resolved MMSE/CD. Therefore, we do not believe that a multi-domain cholesteric structure is present in the unaligned polyfluorene systems considered here.

3) The authors have, in my view, not really addressed the issue of the magnitude of the electric and magnetic dipole moments required in their proposed explanation. They do not come up with estimates for the electric and magnetic dipole moments that lie at the core of their new explanation. The argument provided by the authors that in supramolecular chiral systems also high g -values are observed is invalid because in the supramolecular structures there may be contributions to the circular dichroism from the long-range structural chirality that the authors are trying to disprove. In short, I am not convinced that the magnitudes of the dipole moments required in the new explanation are physically viable.

We apologise for not being clearer in our previous response. Whilst there have been few reports that explicitly attribute such strong dissymmetry (g -) factors solely to natural optical activity, they are not unprecedented either. With careful consideration of the size and relative orientation of molecular transition dipole moments, large dissymmetry factors can be achieved without long-range order effects on the optical response. The dissymmetry factors can be further enhanced through the coupling of dipoles on nearby chromophores. Delocalisation of the excited state over multiple chromophores will result in the breakdown of the dipole approximation, as the extension of the excited state is no longer significantly small than the wavelength of

light. The breakdown of the dipole approximation leads to a larger contribution from the magnetic transition dipoles and electronic quadrupole transition. While the latter is not relevant in the present work, the former will result in increased g -factors. Finally, excited state delocalisation also tends to decrease the electronic transition dipole moment. This will make the electronic and magnetic transition dipole moments closer, again increasing the g -factors.

For example, Schiek *et. al* achieved g -factors approaching 1 *via* molecular design and excitonic coupling (ref 41 in main manuscript, 10.1038/s41467-018-04811-7). In this paper, Mueller matrix spectroscopic ellipsometry and microscopic inspection were used to confirm the absence of any optical effects of long-range structural order. Similarly, g -factors of this order of magnitude have been observed in the circularly polarised emission from cylindrical single-wall carbon nanotubes ($g_{\text{abs}}, g_{\text{lum}} \sim 0.2$), even when the molecules are in solution (DOI: [10.1073/pnas.1717524114](https://doi.org/10.1073/pnas.1717524114)). In this case, the transition dipole moments are as follows: $\mu = 1.25 \times 10^{-18}$ esu cm, $m = 27.7 \times 10^{-18}$ erg G⁻¹, $\theta_{\mu,m} = 180^\circ$.

It is therefore conceivable that a polymer system with appropriately oriented transition dipole moments and exciton coupling may achieve the large g -factors we report. From their response, Referee 3 agrees with our assumptions, recognising the difficulty in getting even an approximate value. We also should insist, as the referee will know, that high chiroptical activity in a supramolecular system does not need, necessarily, to originate from long range effects structural chirality. This is precisely the point of our paper.

We believe that providing estimates for the electric and magnetic dipole moments that lie at the core of this proposal would be highly speculative, because a very precise structural molecular model of our system is not possible at this stage. Very high-level molecular mechanics (which would need state-of-the-art calculations because of the size and complexity of our systems) would be needed to obtain that model. We really feel this is beyond scope for the present time, as it will require a massive further theoretical endeavour.

Reviewer #3 (Remarks to the Author):

The manuscript has greatly improved, thanks to the introduction of carefully conducted and thoughtfully presented thickness-dependent experiments. In the rebuttal, very correctly, the authors put forward a relevant point: “changes in thin film morphology as a function of film thickness”, which may be even more true on account of different spinning rates. Indeed, in general, competing aggregation pathways may lead to polymorphs whose relative weight may be thickness- or spinning rate-dependent. They can be expected to be associated to different CD spectra and the existence of a similar multiplicity of species can be demonstrated by studying CD profiles as a function of the thickness, of the spinning rate, of the distance from the spinning focus. All this would go beyond the present investigation, but I would suggest, as a minor correction to account for this in the text.

We are grateful for this suggestion and are interested in the prospect of studying CD profiles as a function of distance from the spinning focus (particularly pertinent for the design of large area optoelectronic devices). Whilst beyond the scope of this current manuscript, we have previously investigated the CD of our systems as a function of solution concentration and polymer molecular weight; both of which might be expected to modify the polymorphs present within a thin film. We have also previously inspected the CD profiles of variable thickness F8BT:aza[6]H films where spinning rate was used to control film thickness (ref 30 in main manuscript, 21 in SI: 10.1021/acsnano.9b02940). From this work we found no significant changes in the shape of the CD profiles, nor evidence of different polymorphs in AFM or cross-polarised optical microscopy images. We have added the following to the supporting information;

To better understand the origins of this chiroptical response, we controlled the spin-coating speed to fabricate a series of thin films with different thicknesses. In such films, the competition of different aggregation pathways may result in the formation of various polymorphs in principle. However, the variable thickness ACPA films considered here suggest that this is not the case for these materials, with similar shaped CD profiles for thick and thin films and no evidence of different polymorphs in AFM or cross-polarised optical microscopy images.²¹

The point raised by referee 2 is correct in the absence of exciton coupling and indeed even the most extreme cases of intrinsically chiral chromophores (including helicenes and other highly conjugated twisted systems) hardly go over $g=10^2$. In some cases, chiral supramolecular architectures can overcome this limit, thanks to exciton coupling and I consider myself satisfied with the text introduced by the authors.

On account of the value of the work and of the results I'll not insist, recommending further changes, but to me a "unified model" is one, which accounts for several effects in different conditions, which is the opposite of: "Indeed, the aligned scenario, where structural chirality is responsible for the strong CD, is a *totally different mechanism*", or of "***In contrast*** when an alignment layer is introduced, the chiroptical effects in transmission occur due to the creation of a mesoscopic cholesteric-like stack normal to the substrate plane, where the twist and pitch of the polymer layers dictates the size of the chiroptical response through dissymmetric reflection of left- and right-handed CPL."

We apologise for any confusion or frustration caused. While we felt "unified" was a useful term with respect to our study, we have decided not to include it given this further feedback. We have therefore updated the title as follows,

Control of large natural optical activity in π -conjugated polymer thin films

REVIEWERS' COMMENTS

Reviewer #2 (Remarks to the Author):

Report on "Control of large natural optical activity in π n-conjugated polymer thin films"

This report concerns the second revised version.

1) The authors keep dodging the question on the magnitudes of the electric and magnetic transition dipole moments featuring in their modelling. I work this out below. For the fluorene based polymers the electric transition dipole moment μ is on the order of 10 Debye = 10×10^{-18} esu.cm. Using the well-known expression for the degree of circular polarization of natural optical activity:

$$g = 4\text{Im}(\mu \rightarrow \cdot m \rightarrow) / (\mu^2 + m^2)$$

one finds that in order to achieve $g = 1$, one needs a magnetic transition dipole moment $|m|$ that is at least a quarter of the magnitude of the electric transition dipole moment $|\mu|$. So $|m| > 2.5 \times 10^{-18}$ erg G⁻¹ which is equivalent to 270 Bohr magneton. The magnetic dipole moment depends on orbital angular momentum in organic molecules with low spin-orbit coupling and can be approximated by $m = \sqrt{L(L+1)}$ in Bohr magneton, implying that one would need an orbital angular momentum L exceeding 100 units of \hbar ! No magnetic transition dipole moment of such magnitude have been reported as far as I know. The authors in their reply cite a particular report on large magnetic transition dipole moment in molecules, but they failed to notice the correction following the article mentioned (<https://www.pnas.org/content/116/11/5194>). The correct value for the magnetic transition dipole for the circular molecule is 1×10^{-19} erg. G⁻¹, i.e. 25 times smaller than the one needed to explain the high g -values in the fluorene type polymers under study here.

In summary, the authors in their abstract promise a "disruptive mechanistic insight" into the chiroptical properties of π n-conjugated polymers but in fact they regress back to the now century-old molecule-based natural optical activity model and do not provide an account or interpretation of the exceptionally high values for the magnetic transition dipole moments required.

2) The authors seek to rule out the cholesteric model as an explanation for their chiroptical effects. This starts with figure 3 where the experimental data for films without alignment layer are compared to predictions for a monodomain cholesteric film. This is confusing and unnecessary. It is common knowledge that if you take a material with a liquid crystalline order and do not apply an alignment layer that you get a multidomain film. So the fact that the predictions for the monodomain do not fit the experiment are not surprising. This is very ineffective communication where one first bothers the reader with unrealistic scenarios. Later in the text and the SI, the authors then revert their argument that cholesterically ordered materials should show circularly polarized reflection, admitting that in multidomain films such circularly polarized reflection may under certain circumstances be absent from cholesterically organized media. The discussion continues in supporting information and shifts to depolarization rather than circular polarization in reflection as the prime discriminating experimental observable. The discussion becomes increasingly technical in nature, featuring a range of mathematical models that have as far as I can see never been validated on molecular materials. The argument starting with Figure 3 was not that clear and strong to begin with and just fizzles out.

In conclusion, in the title of their contribution the authors promise "Control of large natural optical activity ..." In the main they raise some objections against the interpretation of the large circular polarizations in terms of structural chirality. They propose to go back to the old molecular interpretation but do not account for the magnitude of the transition dipole moments that feature in

natural optical activity model. Materials and optical phenomena have largely been reported on in earlier works. So in the end, the main novelty of the paper seems to be that the authors have reconsidered the natural optical activity model for molecular materials and now propose this as an alternative, phenomenological description of the optical properties of films of polymers. I use "phenomenological" here because the magnetic transition dipole moment in the model is treated as just a fit parameter. So the authors can describe the phenomena with their model but do not demonstrate that it has the power to predict these phenomena starting from molecular structure or organization. I don't think this makes a contribution of markedly outstanding importance to the community working on molecular materials. Then lastly, where is the "control" promised in the title ?

Stefan Meskers

1) The authors keep dodging the question on the magnitudes of the electric and magnetic transition dipole moments featuring in their modelling. I work this out below. For the fluorene based polymers the electric transition dipole moment μ is on the order of 10 Debye = 10×10^{-18} esu.cm. Using the well-known expression for the degree of circular polarization of natural optical activity:

$$g = 4 \operatorname{Im}(\mu \rightarrow \cdot m \rightarrow) / (\mu^2 + m^2)$$

one finds that in order to achieve $g = 1$, one needs a magnetic transition dipole moment $|m|$ that is at least a quarter of the magnitude of the electric transition dipole moment $|\mu|$. So $|m| > 2.5 \times 10^{-18}$ erg G⁻¹ which is equivalent to 270 Bohr magneton. The magnetic dipole moment depends on orbital angular momentum in organic molecules with low spin-orbit coupling and can be approximated by $m = \sqrt{L(L+1)}$ in Bohr magneton, implying that one would need an orbital angular momentum L exceeding 100 units of \hbar ! No magnetic transition dipole moment of such magnitude have been reported as far as I know. The authors in their reply cite a particular report on large magnetic transition dipole moment in molecules, but they failed to notice the correction following the article mentioned (<https://www.pnas.org/content/116/11/5194>). The correct value for the magnetic transition dipole for the circular molecule is 1×10^{-19} erg. G⁻¹, i.e. 25 times smaller than the one needed to explain the high g -values in the fluorene type polymers under study here.

In summary, the authors in their abstract promise a “disruptive mechanistic insight” into the chiroptical properties of π -conjugated polymers but in fact they regress back to the now century-old molecule-based natural optical activity model and do not provide an account or interpretation of the exceptionally high values for the magnetic transition dipole moments required.

We believe the reviewer has not understood our response and we therefore apologise if we were not clear. In their previous comments, Reviewer 2 requested examples of the magnitudes of the dipole moments in systems where long-range structural chirality is not responsible for strong g -factors. We therefore provided examples of such cases in small molecules (10.1038/s41467-018-04811-7: $g_{\text{abs}} = 0.75$, 10.1073/pnas.1717524114: $g_{\text{abs}} = 0.17$); although we note our manuscript represents the first explicit assignment of natural optical activity being proposed as the origin of the strong chiroptical effect in π -conjugated polymers. The reason we “dodged” assigning a value to the electric and magnetic transition dipole moment, is that the analysis provided by the reviewer above is not appropriate. We do not claim that the origin of the natural optical activity observed is “molecule-based”. Indeed, we agree that the “well-known” expression for g -factor cited above gives values for the magnetic transition dipole moment that are unrealistic. This expression for g -factor (i) assumes isolated, non-interacting chromophores, (ii) assumes molecules that are much smaller than the wavelength of light and (iii) neglects anisotropic effects (i.e. it does not consider quadrupole terms or higher order coupling). For

Figure 1: Absorption coefficient of as-cast and annealed F8BT:aza[6]H thin films.

anisotropic systems such as the polymer-based thin films considered here, a more sophisticated expression is required (10.1021/j100632a012, 10.1021/j100632a012), and the coupling of transition dipole moments on nearby polymer chains must be considered.

Taken together, these factors can all result in an increase in the intrinsic *g*-factor of a material and do not require an unrealistic magnetic transition dipole moment as would be expected for an isolated chromophore. As we mentioned in our previous response, to provide a more quantitative analysis of our polymeric system with appropriately oriented transition dipole moments and exciton coupling would require a very precise structural model, which would need high-level quantum chemical methods potentially invoking coarse-grained approximations that have not previously been attempted and are beyond the scope of the current work.

We should emphasise that Reviewer 2 does not provide any evidence that there is structural chirality in these systems – which is the principle mechanism assigned in the literature - or propose an alternative universal description that could explain all our experimental results.

At the request of the reviewer, we have updated the manuscript (page 16, new text in blue below) to include further details of this discussion about transition dipole moments and will continue our efforts to identify a more precise structural model.

The very large chiroptical effects associated with natural optical activity that we report contrasts with the small effects typically observed for small molecules in dilute solution, but is reminiscent of the high optical activity of chiral aggregates where supramolecular chirality leads to CD effects that are magnitudes higher than those of the isolated molecules.^{54–57} Dissymmetry factors of isolated chromophores can be enhanced through the excitonic coupling of dipoles on nearby chromophores, evidence of which lies in the CD spectra of annealed ACPCA systems (Figure 2 and 7). Delocalisation of the excited state over multiple chromophores will also result in the breakdown of the dipole approximation, as the extended excited state is no longer significantly smaller than the wavelength of light. Finally, for unaligned ACPCA films, the in-plane helical periodic modulation of the polymer in the double twist cylinder blue phase may give rise to a chromophore large enough to possess no local symmetry, which leads to a greater contribution from the magnetic transition dipoles and electronic quadrupole transition to the dissymmetric response.⁵⁸

We have also included the following text in the Supporting Information (page 84),

We should note that the large g -factors observed for these systems contrast the small effects typically seen for small molecules. In the case of small molecules, the g -factor can be calculated from²⁷;

$$g = 4 \frac{R}{D} = 4 \times \frac{Im \boldsymbol{\mu}_{ij} \cdot \mathbf{m}_{ij}}{|\boldsymbol{\mu}_{ij}|^2 + |\mathbf{m}_{ij}|^2} \quad \text{Equation S10}$$

Where $\boldsymbol{\mu}$ is the electric transition dipole moment, \mathbf{m} the magnetic transition dipole moment, R the rotational strength and D the dipole strength of an electronic transition from i - j . Based on Equation S10 and the large $\boldsymbol{\mu}$ s of fluorene polymers, g -factors > 1 would require unrealistically large magnetic transition dipole moments. However, the well-known expression for g -factor (i) assumes isolated, non-interacting chromophores, (ii) assumes molecules that are much smaller than the wavelength of light and (iii) neglects anisotropic effects (i.e. it does not consider quadrupole terms or higher order coupling). For anisotropic systems such as the polymer-based thin films considered here, a more sophisticated expression is required, and the coupling of transition dipole moments on nearby polymer chains must be considered.²⁸

2) The authors seek to rule out the cholesteric model as an explanation for their chiroptical effects. This starts with figure 3 where the experimental data for films without alignment layer are compared to predictions for a monodomain cholesteric film. This is confusing and unnecessary. It is common knowledge that if you take a material with a liquid crystalline order and do not apply an alignment layer that you get a multidomain film. So the fact that the predictions for the monodomain do not fit the experiment are not surprising. This is very ineffective communication where one first bothers the reader with unrealistic scenarios. Later in the text and the SI, the authors then revert their argument that cholesterically order materials should show circularly polarized reflection, admitting that in multidomain films such circularly polarized reflection may under certain circumstances be absent from cholesterically organized media. The discussion continues in supporting information and shifts to depolarization rather than circular polarization in reflection as the prime discriminating experimental observable. The discussion becomes increasingly technical in nature, featuring a range of mathematic models that have as far as I can see never been validated on molecular materials. The argument starting with Figure 3 was not that clear and strong to being with and just fizzles out.

We respectfully disagree with the reviewer that we 'bother the reader with unrealistic scenarios'. Our discussion of mono- and multi-domain cholesteric stacks are introduced in the same paragraph (page 8), rather than separated as the reviewer suggests. Whilst

the formation of a multi-domain cholesteric stack in a liquid crystalline polymer may be “common knowledge” to the reviewer, the same cannot be said for the broader scientific community, who frequently use the spectroscopic results obtained using alignment layers to describe the molecular packing of thin films where alignment layers have not been used (see Table S1 for an overview of the mechanisms used to explain chiroptical phenomena). Beyond comparing the predicted circular reflectance spectra of a mono-domain cholesteric stack with our experimental results (Figure 3, Figure S7), we mention the two short-comings of both a coherent and incoherent multi-domain models in the same paragraph (the circular terms in reflectance are not cancelled at multiple angles of incidence and the introduction of depolarisation, which we don't see in our data, Figure S8).

Technical details of both the coherent and incoherent multi-domain models were provided **at the request of the reviewer** to allow readers to reproduce the simulations if necessary. Our initial concerns with generating such an incoherent multi-domain model was the lack of an experimental control sample to validate it, but similar mathematical approaches have been previously explored by M.-Y. Xie *et al.* (DOI: 10.1103/PhysRevB.90.195306) when considering InN films with mixed cubic zincblende and hexagonal wurzite crystallites. The multi-domain cholesteric models are based on the weighted sum of the Jones matrices (for the coherent case) or Mueller matrices (for the incoherent case) of several cholesteric grains with different starting orientations. The outputs of these multi-domain models are essential for us to disprove that structural chirality is the origin of the strong chiroptical effects seen in these films, and as such we believe that the Supporting Information of a manuscript such as this is the appropriate place to provide technical details of the models. We will, of course, be guided by the editorial team.

To make it clearer to the reader, we have adjusted the aforementioned paragraph as follows (new text in blue),

We first attempted to model the MMSE data by assuming the chiroptical effects arose from **either a mono- or multi-domain** cholesteric stack structure. As the refractive index of these materials is ≈ 2 at wavelengths close to $\lambda_{\text{CD Max}}$, the pitch **of a mono-domain cholesteric** would be ≈ 250 nm to generate the maximal chiroptical effect observed (Figure S7). In transmission, this model of structural chirality produces linear effects in several MM elements that are not exhibited in experimental data (Figure S7), whilst in reflection, the **monodomain** cholesteric stack model produces strong differential reflection of CPL that does not appear in our measurements (Figure 3, red curves). These findings show that the origins of the CB and CD in non-aligned chiral films do not result from a mesoscopic model based on a dielectric tensor that is a periodic function of the thickness, as would occur in structurally chiral monodomain cholesteric stack systems. We shall refer to this

mesoscopic description as the twisted dielectric tensor model. We next considered helical multi-domain models,⁴³ which evaluate the optical response of incoherent (domain sizes $>$ the wavelength of light) and coherent (domains sizes \leq the wavelength of light) superpositions of cholesteric grains (Figure S8 and accompanying discussion). Neither an incoherent or coherent multi-domain model can satisfy the MMSE results acquired at multiple angles of incidence or without introducing significant depolarisation, which is not observed in our measurements. The uniformity of the circular response was evaluated using spatially resolved MMSE (Figure S8), which, comparable to the spatially resolved CD measurements described above (Figure S2), show no evidence of a multi-domain structure.

In conclusion, in the title of their contribution the authors promise “Control of large natural optical activity ...” In the main they raise some objections against the interpretation of the large circular polarizations in terms of structural chirality. They propose to go back to the old molecular interpretation but do not account for the magnitude of the transition dipole moments that feature in natural optical activity model.

We have addressed the issue of transition dipole moments in the answer above, in the main manuscript and also in our previous responses to the reviewer.

Materials and optical phenomena have largely been reported on in earlier works.

This statement is incorrect. This manuscript compares several different polymeric systems (F8BT, PFO, F8T2:aza[6]H and cPFBT, cPFO), not all of which have been reported in earlier works and none of which have been studied in such detail. *In situ* details on the formation of the chiral phases, as well as an optical and molecular model that can describe all experimental results acquired in reflection and transmission, has never before been reported. Further, the manuscript looks to compare the fundamental mechanisms by which the optical phenomena manifest, not to introduce novel polymers/molecules.

So in the end, the main novelty of the paper seems to be that the authors have reconsidered the natural optical activity model for molecular materials and now propose this as an alternative, phenomenological description of the optical properties of films of polymers. I use “phenomenological” here because the magnetic transition dipole moment in the model is treated as just a fit parameter. So the authors can describe the phenomena with their model but do not demonstrate that it has the power to predict these phenomena starting from molecular structure or organization.

Once again, we disagree with this statement. In the manuscript we show that starting from the thickness and optical constants of a polymer-based chiral film, a model based on magneto-electric coupling can be used to predict the chiroptical phenomena recorded in both reflection and transmission for a range of π -conjugated systems (Figure 2 and Figure S5). This is a significant result for the field, where

models based on the evaluation of aligned polymer films have previously been used to justify the strong chiroptical response.

I don't think this makes a contribution of markedly outstanding importance to the community working on molecular materials. Then lastly, where is the "control" promised in the title ?

Whilst the reviewer may believe that this will not "a contribution of markedly outstanding importance to the community working on molecular materials", the data indicate otherwise. We note that the pre-print has been well received by the scientific community (<https://www.altmetric.com/details/81846178>), is in the 'top 5% of all research outputs scored by Altmetric' and has a 'High Attention Score compared to outputs of the same age and source (96th percentile)', suggesting high interest.

The title was changed in the review process since a previous reviewer did not agree with our previous title. In our revised title, the *control* of natural optical activity is achieved through the incorporation of a chiral additive and appropriate post-deposition processing (i.e. thermal annealing). However, we are happy to change the title again, if this is misleading. We have updated the title as follows:

Natural optical activity as the origin of the large chiroptical properties in π -conjugated polymer thin films